# Microbial and metabolic succession on common building materials under high humidity conditions

Simon Lax[1,9], Cesar Cardona [2], Dan Zhao[3], Valerie J. Winton [4], Gabriel Goodney [5], Peng Gao[4], Neil Gottel[6], Erica M. Hartmann[7], Chris Henry[8], Paul M. Thomas [4], Scott T. Kelley[5], Brent Stephens[3] & Jack A. Gilbert [6]

Despite considerable efforts to characterize the microbial ecology of the built environment, the metabolic mechanisms underpinning microbial colonization and successional dynamics remain unclear, particularly at high moisture conditions. Here, we applied bacterial/viral particle counting, qPCR, amplicon sequencing of the genes encoding 16S and ITS rRNA, and metabolomics to longitudinally characterize the ecological dynamics of four common building materials maintained at high humidity. We varied the natural inoculum provided to each material and wet half of the samples to simulate a potable water leak. Wetted materials had higher growth rates and lower alpha diversity compared to non-wetted materials, and wetting described the majority of the variance in bacterial, fungal, and metabolite structure. Inoculation location was weakly associated with bacterial and fungal beta diversity. Material type influenced bacterial and viral particle abundance and bacterial and metabolic (but not fungal) diversity. Metabolites indicative of microbial activity were identified, and they too differed by material.

[1] Department of Ecology and Evolution, The University of Chicago, 1101 E 57th St, Chicago, IL 60637, USA. [2] Graduate Program in Biophysical Sciences, The University of Chicago, 924 E. 57th Street, Chicago, IL 60637, USA. [3] Department of Civil, Architectural, and Environmental Engineering, Illinois Institute of Technology, 3201 S Dearborn St, Chicago, IL 60616, USA. [4] Proteomics Center of Excellence and Department of Molecular Biosciences, Northwestern University, 710 N. Fairbanks Ct, Chicago, IL 60611, USA. [5] Department of Biology, San Diego State University, 710 N. Fairbanks Ct, San Diego, CA 92182, USA. [6] Department of Pediatrics and Scripps Institution of Oceanography, University of California San Diego, La Jolla, CA 92037, USA. [7] Department of Civil and Environmental Engineering, Northwestern University, 2145 Sheridan Road, Evanston, IL 60208, USA. [8] Mathematics and Computer Science, Argonne National Laboratory, 9700S. Cass Avenue, Lemont, IL 60439, USA. [9] Present address: Center for the Physics of Living Systems, Department of Physics, MIT, 400 Technology Square, Cambridge, MA 02139, USA. These authors contributed equally: Simon Lax, Cesar Cardona. Correspondence and requests for materials should be addressed to B.S. (email: brent@iit.edu) or to J.A.G. (email: gilbertjack@gmail.com)

The microbiology of the built environment (BE) comprises bacteria, archaea, fungi, viruses and protists, all of which maintain growth potentials under varying physicochemical regimes. Many recent studies of this ecosystem have applied molecular sequencing techniques to characterize microbial community dynamics and their relationship to occupant density, material type, location, and environmental conditions[1–5]. However, most of these studies have investigated communities sampled from relatively dry materials on which microbes are likely biologically inactive unless they experience liquid water or high relative humidity[3] (RH). It is widely accepted that fungal growth can occur at RH > 75–80% and material decay can occur at RH > 95%, depending on material[6,7].

Dampness is a fairly common occurrence in buildings, with approximately half of all homes in the U.S. having experienced dampness or mold[8]. Building material dampness can originate from many sources, including bulk liquid entry from floods, extreme weather events, and plumbing system problems; rain or snow entry through leaks in building envelopes and roofing systems; and high water vapor content resulting from moisture migration through building materials or condensation of warm humid air on cold surfaces[9]. Dampness and the presence of visible mold have been consistently associated with adverse human health outcomes, including respiratory and allergic effects[10–13]. These associations may result from a combination of exposure to specific microbial agents[14], varied gene expression and metabolism[15], and the release of fungal metabolites including mycotoxins[16] and microbial volatile organic compounds[17].

Although fungal growth on building materials has been studied for decades[18–21], only a limited number of studies have used molecular techniques to investigate bacterial and fungal growth, microbial community dynamics, and metabolic activity on common buildings materials exposed to liquid water and/or high humidity conditions[22]. Therefore, we characterized the bacterial and fungal concentration and diversity, as well as the production of microbial metabolites, on samples of four common building materials incubated at constant ~94% relative humidity: oriented strand board (OSB), medium density fiberboard (MDF), gypsum wallboard, and mold-resistant (i.e., mold-free, or "MF") gypsum wallboard. The materials were selected to represent a relatively wide variety of common building and furniture material types that were also likely to experience high variability in microbial growth. We varied the BE source of inoculation and wet half of the samples to assess how indoor microbial sources and the presence of liquid water influence community structure and metabolite profiles of these materials over time. We used several techniques to quantify microbial growth, microbial community composition, and functional metabolism including: bacterial- and viral-like particle counts, image processing of visible mold growth, qPCR, amplicon sequencing of 16S and ITS rRNA marker genes, and metabolomics. Results from these different methods were integrated via co-occurrence network approaches, which provided insights into microbial community organization and environmental interaction. Improved understanding of how bacterial and fungal metabolism is shaped by environmental properties (e.g., the presence of water, surface material composition) and inoculating source (e.g., building location, occupancy patterns) could have important implications for building design, construction, and management, and potentially for occupant health, such that determining the microbial dynamics in these high RH environments should be an important research priority.

## Results

**Experimental setup**. Multiple sampling strategies were tested, including repeated sampling of the same coupons at each time point and sampling new coupons at each time point (Supplementary Fig. 1). After statistical verification of these two approaches, all samples of the same type were combined as technical replicates (Supplementary Tables 2 and 3). Microbial datasets were later rarefied to an even sequencing depth: 1000 reads for bacteria and 10,000 reads for fungi. Unfortunately, rarefaction removed all bacterial samples from MDF materials, which had very low read counts. After rarefying the data, a comparison of the control (lab-inoculated) and noncontrol (residence-inoculated) samples reflected that control samples looked very similar in bacterial and fungi diversity to noncontrol samples (Mantel ≥ 0.49 and ≥0.43 for Location 1 and Location 2 respectively, all with $p < 1E-05$), perhaps because air could still transmit through the nonhermetic foil cover and microbes from the interior of the wood (not killed with the sterilization) could have found their way to the surface. It is also possible that the coupon itself could remain an important reservoir of microbial communities that contribute to the microbial diversity of the samples. Based on these results, the covered laboratory location was treated similarly to the other two locations. For more details see the section "Methods".

**Visible growth, particulate counts and qPCR**. Visible microbial growth occurred much faster and covered a far greater percentage of the surface area on wet coupons than on nonwetted coupons (Fig. 1a). OSB and MDF had the greatest coverage and fastest growth: all wet OSB and MDF coupons reached at least 50% visible microbial coverage by day 20, while nonwetted coupons of these types reached <25% coverage. No growth was ever visible on the mold-resistant gypsum coupons. Epifluorescence microscopy revealed that counts of bacterial-like particles (BLP) and viral-like particles (VLPs) calculated on samples 0–15 days post incubation were strongly correlated ($R^2 = 0.65$, $p = 2.8e^{-23}$) (Fig. 1b), with VLP counts statistically lower than BLP counts in all samples (ANOVA ≤ $10^{-4}$) and in both wet and nonwet conditions (two-sided nonparametric $t$ test $p \leq 0.035$) (Fig. 1c). This is in keeping with previous research that found very low VLP:BLP ratios on built surfaces[23] and indoor aerosol samples[24]. In our dataset, the mean log(VLP)-log(BLP) ratio was 0.86 (SD = 0.07, $N = 96$), with a minimum of 0.61 and a maximum of 1.02.

While BLP (only estimated for day 0 to day 15 samples) and bacterial qPCR agree that wetted samples had higher counts than nonwetted samples, with a 4 and 99-fold median cell count increase respectively, cell counts inferred from these two methods drastically differ between material types and over time. Most notably, MF-gypsum had the highest BLP counts but also the lowest 16S rRNA qPCR median cell counts (44 to 97-fold lower than other materials). Moreover, the BLP cell counts were essentially constant over time, while qPCR counts steadily increased, with 30 days post incubation being 209-fold greater than the 394 cells median count per μL at day 0. To further confirm the differences, we calculated the overall correlation between paired bacterial qPCR and BLP counts and the results were not significant, emphasizing different biases for each method.

For fungal qPCR we observed MF-gypsum had the lowest abundance, while all other materials had a range of 37- to 239-fold increase over MF-gypsum. Wetted samples revealed a 72-fold increase in qPCR median read abundance over nonwetted samples. Also, the qPCR read abundance increased steadily over time, in such a way that day 30 was 750-fold greater than the five cells median count per μL at day 0.

**Bacterial, fungal and metabolite diversity**. The bacterial and fungal communities in our study tended to decrease in diversity

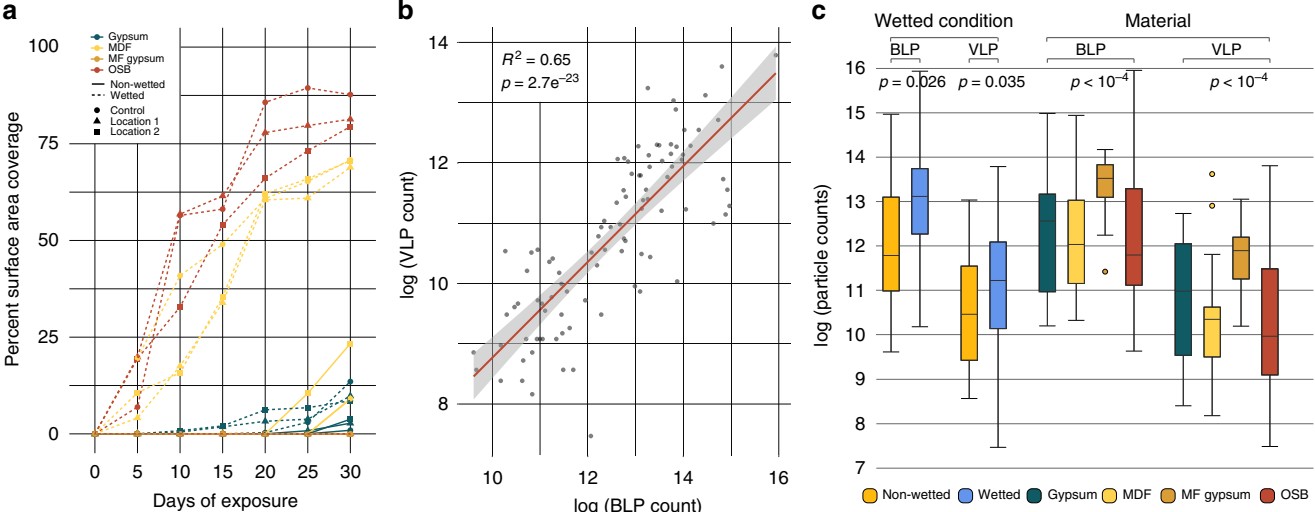

**Fig. 1** Microbial growth rates vary across sample types. **a** Percent of surface area covered by visible microbial growth through time ($n = 168$, 4 materials, 3 locations, 2 wetting conditions, 7 time points). Color indicates coupon material, point shape indicates inoculating location, and line type indicates whether the coupon was wet before incubation. **b** Correlation in the counts of bacteria-like (BLP) and viral-like particles (VLP) across all coupons ($n = 96$ samples, 4 materials, 3 locations, 2 wetting conditions, 4 time points). **c** Boxplots of BLP and VLP counts by wetting condition and by material ($n = 96$ samples). Box boundaries correspond to the first and third quartiles and whiskers extend to the largest values no further than 1.5 times the distance between the first and third quartiles. Source data are provided as a Source Data file

over time, as measured by the Shannon Index (Shannon H′), which incorporates both the richness and evenness of the community. Given that our data were rarified to an even depth before analysis, this decrease in diversity is indicative of the increasing relative abundance of certain community members, and suggests the preferential proliferation of certain taxa in the inoculating community. In our bacterial dataset, wetted samples experienced faster declines in diversity than nonwetted samples, and were significantly lower in diversity at the end of the study than nonwetted samples (Fig. 2a), suggesting that certain bacterial taxa grew quickly in the wet environment and became dominant within the community. In our ITS dataset, we also observed a faster decline in diversity in wetted samples, although wetted samples were significantly more diverse than nonwetted samples by the end of the study (Fig. 2b). The decrease in fungal diversity in wetted samples was not monotonic, with an initially steep decline and a subsequent increase. This may reflect fast growth by a small number of taxa that quickly dominated the community, followed by the growth of other taxa with slower growth rates. Similar patterns occurred within the diversity changes for each individual material (Supplementary Figure 2) with the exception of a lack of bacterial growth for wet MDF samples and reduced bacterial growth on dry OSB after the study was half way completed. In contrast, we observed no significant changes in the metabolic diversity over time for either wetted or nonwetted samples (Fig. 2c).

**Microbial compositional changes**. Across all samples, the diversity of bacteria within the community was significantly correlated to the diversity of fungi (Pearson $\rho = 0.28$, $p = 0.0003$) (Fig. 3a). Interestingly, neither bacterial nor fungal diversity was significantly correlated to the metabolite diversity. We observed striking changes in the relative abundance of certain bacterial (Fig. 3b) and fungal (Fig. 3c) genera over time, which were largely dependent on wetting condition. In the bacterial dataset, *Bacillus* almost immediately came to dominate wet samples, with an average relative abundance as high as 50% after 10 days, even though it represented a negligible part of the community at the start of sampling. *Bacillus* abundance also increased in nonwetted

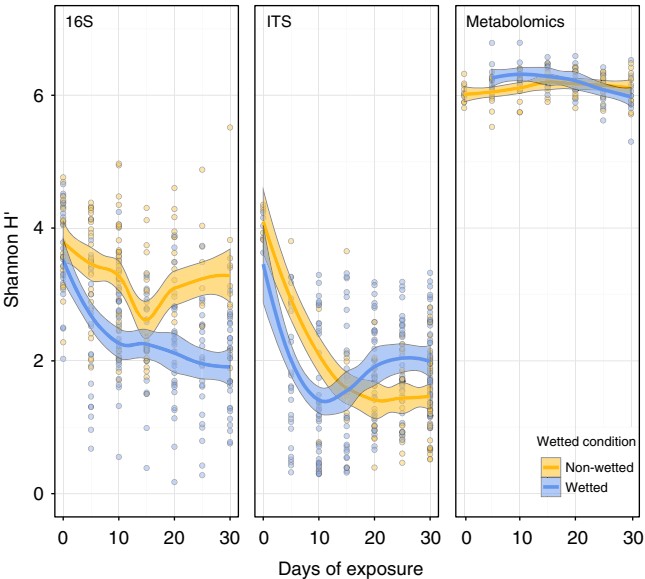

**Fig. 2** Change in the Shannon Index of samples over time. Points represent individual samples and the trend lines are a smoothed moving-average of the mean and shaded regions indicate the standard error ($n = 338$, 330, and 144 samples for 16S, ITS, and Metabolomics, respectively)

samples, although to a much smaller extent. A similar pattern was observed for the genera *Pseudomonas* and *Erwinia*, which also represented a very small fraction of community diversity at the start of sampling but quickly increased in abundance in wet (but not nonwetted) samples. Interestingly, a large percentage of reads from early time point samples, both wet and nonwet, were of chloroplast origin. In wet samples, the number of chloroplast reads quickly declined as bacterial genera proliferated. In non-wetted samples, chloroplast read abundance remained high, and dominated the sequencing effort to such an extent that discarding those reads would have dropped the majority of nonwetted samples below the rarefaction depth. While these likely represent

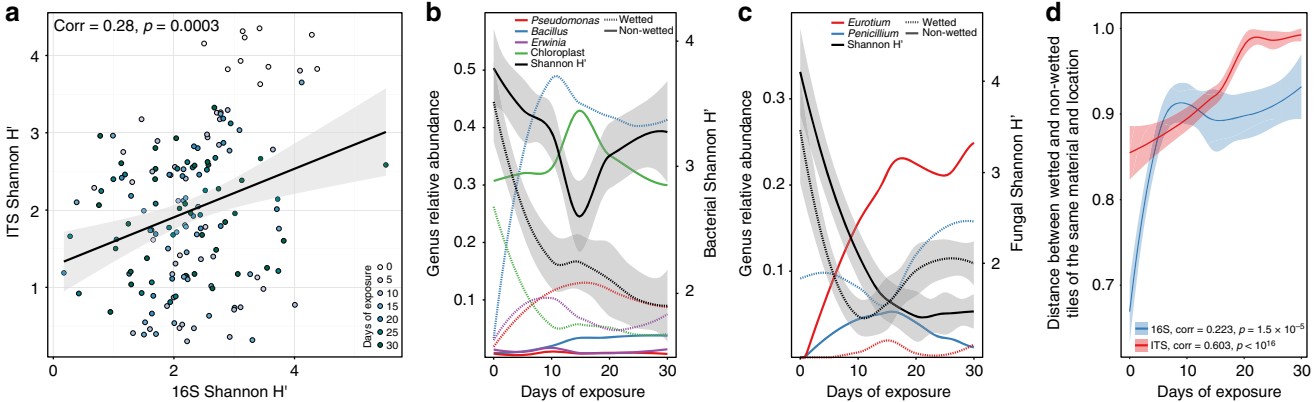

**Fig. 3** Overview of community succession. **a** Fungal diversity and bacterial diversity are significantly correlated across communities ($n = 153$ samples). Points represent individual samples, colored by the time point at which the sample was taken. **b** Changes in the relative abundance of selected bacterial genera over the course of succession ($n = 338$ samples). Lines represent a smoothed moving-average of the mean. Genus is indicated by color and wetting condition is indicted by line style. Average community diversity (Shannon H′ at OTU level, as in Fig. 3) is indicated by black lines with standard error indicated by the gray-shaded region. Genus abundance is indicated on the left $y$-axis and Shannon H′ is indicated on the right $y$-axis. **c** Changes in the relative abundance of selected fungal genera over the course of succession ($n = 330$ samples). Formatting is as in **b**. **d** Wet vs. nonwetted replicates of coupons of the same material and inoculating location become increasingly dissimilar over the course of community succession ($n = 338$, 330 samples for 16S and ITS, respectively). The $y$-axis is the Bray−Curtis distance between replicates. Spearman correlation between community dissimilarity and time is indicated in the legend error

residual DNA signatures from the plant material used to construct each coupon, we have chosen to keep them in the analysis. Supplementary Fig. 3A shows how chloroplast sequence abundance varies by wetted condition and material type.

The majority of reads in the ITS dataset that could be taxonomically assigned to a genus belonged to one of two genera: *Eurotium* and *Penicillium*. *Eurotium* abundance was negligible at the beginning of community succession but quickly flourished in nonwetted samples, becoming the most abundant known genus in those samples within 10 days (Fig. 3c). By contrast, *Eurotium* was not abundant in wet samples. *Penicillium* abundance was, on average, consistently higher in wet samples than in nonwetted samples, and its abundance was significantly anticorrelated to *Eurotium* relative abundance (Pearson $\rho = -0.12$, $p = 0.033$). These taxa-specific changes were mirrored by community-level differentiation, where wet vs. nonwetted coupons of the same material and inoculating location became significantly more dissimilar (Bray−Curtis, Spearman's correlation, $p < 0.01$) in both their bacterial and fungal community structure over time (Fig. 3d). Supplementary Fig. 3B shows how this dynamic varies by material type.

**Factors associated with microbial and metabolite diversity.** We used ANOSIM to determine the factors significantly correlated with differences in the microbial communities across our three datasets. Bray−Curtis dissimilarity was calculated for the bacterial, fungal, and metabolite datasets, and ANOSIM was used to determine whether distances between samples of the same metadata factor (i.e. wetting condition, inoculating location, and material) were significantly lower than distances between samples of different types (Supplementary Fig. 4). In our bacterial dataset, wetting condition, location, and material each had a significant impact on bacterial community structure (all $p < 0.0001$ based on $10^5$ randomized permutations), with wetting having the most pronounced effect ($R = 0.418$). Generally, nonwetted samples tended to be more similar to each other than wet samples were to each other, which is likely due to the dominance of a single chloroplast OTU. Material had a less pronounced effect ($R = 0.247$) and location had the least evident effect on bacterial community structure ($R = 0.133$).

Interestingly, fungal community structure was not significantly described by variance in material, while location had a relatively weak ($R = 0.129$) though highly significant ($p < 0.0001$) association, suggesting that variations in fungal communities that settle on materials (which have been shown to be driven largely by outdoor fungal communities[25]) influence community structure upon experiencing wetting and high RH conditions. Wetting condition was by far the most influential factor influencing fungal community structure ($R = 0.564$, $p < 0.0001$), and in contrast to the bacterial data, wet samples were much more similar to each other than were nonwetted samples. Metabolite diversity within the community was also affected by wetting condition ($R = 0.276$, $p < 0.0001$), with nonwetted samples more similar to each other than wet samples. Material also played a significant role in metabolite diversity ($R = 0.231$, $p < 0.0001$), and mold-free gypsum samples were particularly metabolically similar, likely due to the lack of fungal growth and the underlying chemical composition of the material. Inoculating location had no significant effect on the diversity of metabolites despite having a significant effect on both the bacterial and fungal community membership. We visualized sample similarity using nonmetric multidimensional scaling (NMDS) ordination based on Bray −Curtis dissimilarity (Fig. 4). We converted material, location, and wetting condition into binary variables ($1 = $ yes, $0 = $ no), which were fit onto the ordination, keeping only the significant vectors ($R^2$ values for each vector and their significance is presented in Supplementary Table 1). Visually, both bacterial and fungal beta diversity were more differentiated by wetting condition, while metabolites were visually differentiated by both wetting condition and surface material, likely due to the underlying chemistry of the material and then the subsequent metabolic activity of the microbes when coupons were wetted.

**Bacterial and fungal network co-occurrence.** SparCC[26], an algorithm developed to quantify correlations on compositional microbial abundance data (data that has been subject to rarefaction), and applied with correlation threshold >0.4, uncovered co-occurrence patterns between taxa from each kingdom. In the bacterial network (Supplementary Fig. 5) three co-occurrence clusters were identified, the *Bacillus* cluster, *Pseudomonas* cluster,

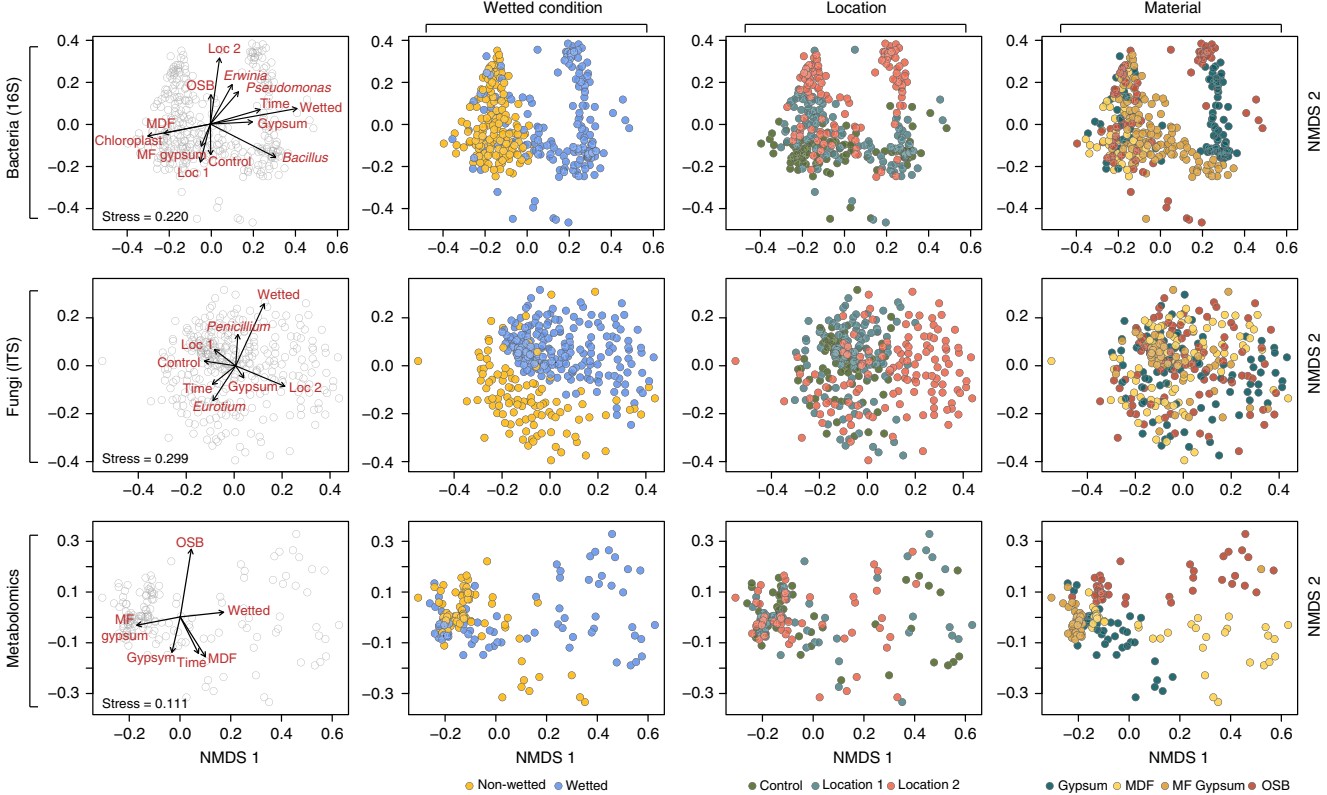

**Fig. 4** NMDS plots illustrate clustering of sample diversity by sample type. Each row comprises four identical NMDS plots ($n = 338$, 330, and 144 samples for 16S, ITS, and Metabolomics, respectively). The leftmost plot illustrates the ordination's association with environmental variables and the remaining plots color sample points by various metadata factors. The stress on the NMDS plot is indicated in the rightmost plot in each row error. NMDS non-metric multi-dimensional scaling

and a cluster comprising chloroplasts and mitochondria. As expected these groups correspond with the most abundant taxa. On all wet materials and on all samples of gypsum (both wet and nonwet), 95% of associations between *Bacillus* and *Pseudomonas* were negative correlations (Supplementary Figs. 6 and 7). On nonwetted OSB, MDF and MF-Gypsum there were no negative correlations between *Pseudomonas* and *Bacillus*. Interestingly, there is a dramatic increase in the absolute number of significant co-occurrence relationships between bacterial OTUs in wet (74) vs. nonwetted samples (48), which is a 54% increase in the number of edges, suggesting that the wetting event has made the environment more suitable for microbial growth interactions. In the fungal correlation network, *Penicillium* OTUs co-occurred with many unknown fungal genera, while OTUs corresponding to *Aspergillus* and its subset, *Eurotium*, maintained monophyletic clusters (Supplementary Fig. 8). As with the bacterial co-occurrence networks, fungal OTUs associated with wet coupons had negative correlations among each other, although the number was much smaller than for bacteria. Only seven fungal OTUs were negatively correlated on wet materials, mainly between unknown genera and an abundant *Penicillium* OTU (Supplementary Fig. 9). Strikingly, unlike bacteria, the absolute number of significant co-occurrence relationships between fungal OTUs declined in wet (555) vs. nonwetted samples (1133), suggesting an inverse co-abundance response between bacteria and fungi during growth.

To better understand the co-associations between bacteria and fungi, 16S and ITS OTUs were co-correlated in a single network. A random walk-based method uncovered four distinct modules within the network, with a modularity of 0.45 (Fig. 5a). In general, the taxa present in each sample tended to cluster within an individual network module (median sample association to

module = 0.88). We correlated various metadata factors to module membership (Fig. 5b) and observed that wetting condition had a significant impact on which samples dominated each module: modules 1 and 3 were associated with wet samples, while modules 2 and 4 were associated with nonwetted samples (Fig. 5c). Location 1 samples dominated module 3, while Location 2 samples dominated module 1 (Fig. 5d). Overall, wetting condition appears to be the most important factor driving community succession, resulting in two different community structures even when the source community is identical. We also visualized the nodes that were assigned to the genera previously discussed in Fig. 3. Nodes in the bacterial genera *Bacillus, Pseudomonas*, and *Erwinia*, as well as the fungal genus *Penicillium*, were nearly exclusively clustered in the two wet-associated modules (1 and 3), while chloroplast reads and *Eurotium* nodes all clustered within the nonwetted modules (2 and 4; Fig. 5e).

**Metabolite network co-occurrence**. A co-occurrence network correlation was calculated for the sample metabolite profiles (Fig. 6). As these data are not compositional, we built this network using significantly positive Spearman correlations between nodes and included only the 1000 most abundant metabolites in the dataset. This resulted in a network with 149,316 edges (density = 0.30) when the significance threshold (alpha) was set to 0.001. Using the same module discovery method described above, we uncovered seven distinct modules (modularity = 0.32), excluding 12 metabolites around the periphery of the network that clustered into modules of <5 nodes. Three modules (3, 4, and 7) were significantly correlated with wet samples, while modules 1, 2, 5, and 6 were associated with nonwetted samples. There was almost no correlation between network modules and

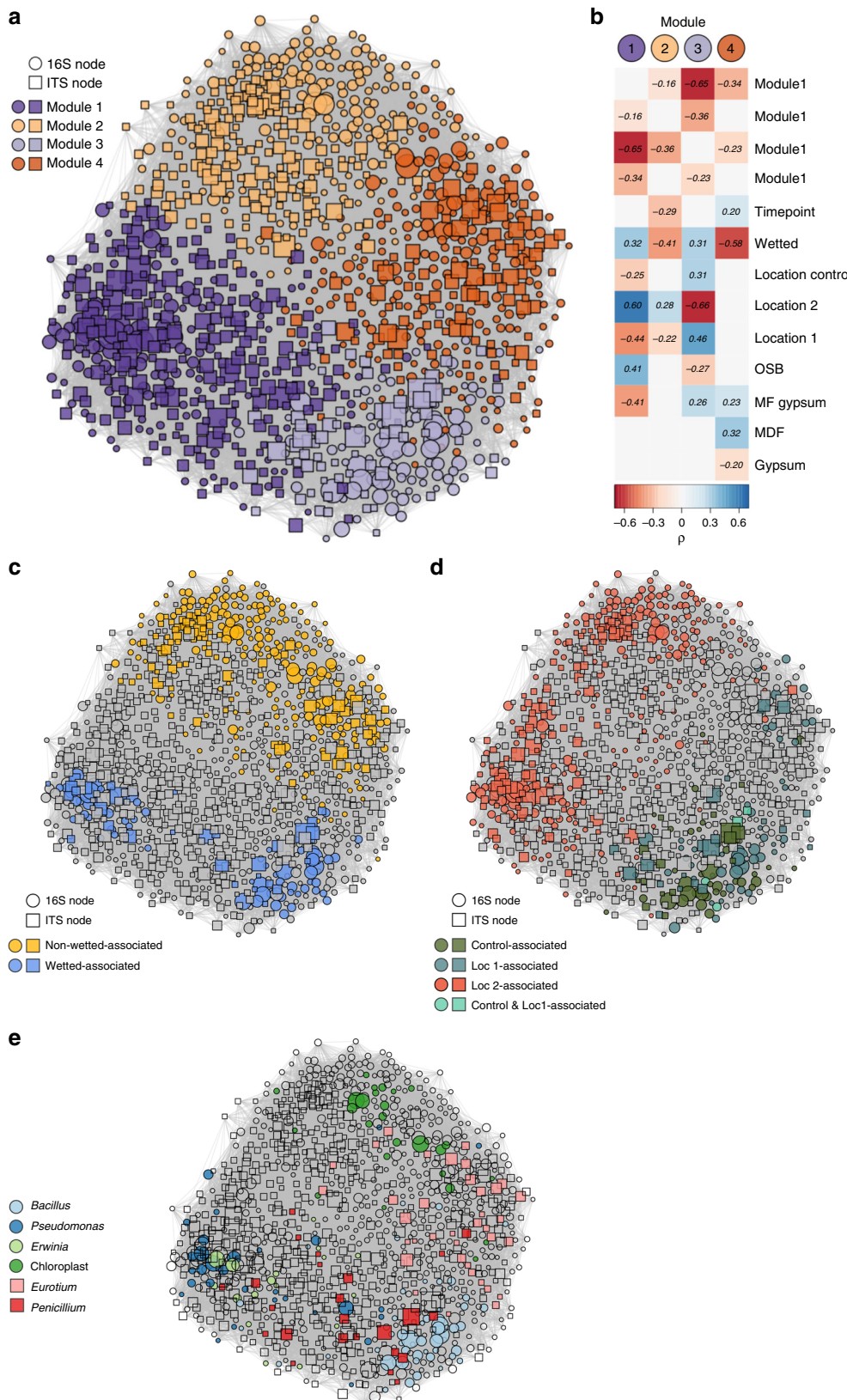

**Fig. 5** Network of SparCC OTU correlations. **a** Edge-weighted, spring-embedded network ordination, with nodes colored by module membership. Node shape represents node type (16S or ITS) and node size is based on the log-transformed abundance of each node ($n = 153$ with both 16S and ITS, respectively). **b** Correlations between metadata factors (treated as dummy variables where true = 1 and false = 0) and the percent of reads in network modules. Nonsignificant correlations are not shown. **c** Taxa enriched in wet or nonwetted samples, as determined through a two-sided nonparametric *t* test with $10^5$ permutations. **d** Taxa enriched in samples originating from an individual inoculating location, with statistical methods as in **a**. **e** Taxonomy of nodes in the genera included in Fig. 4

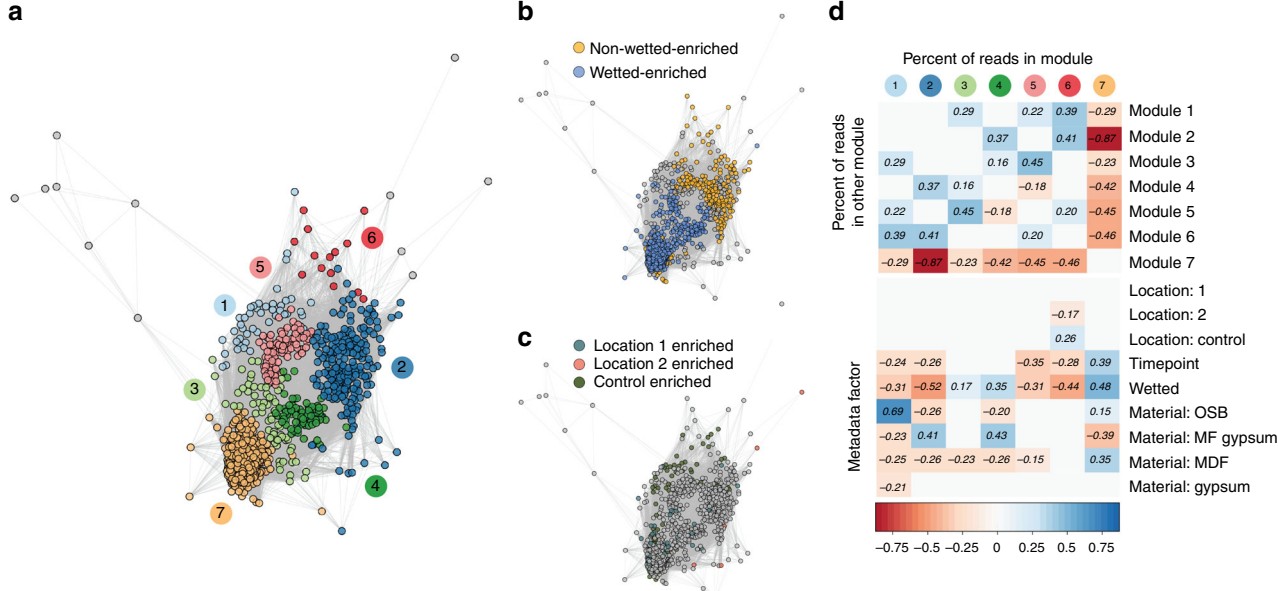

**Fig. 6** Metabolite co-occurrence network. **a** Network of significantly positive Spearman correlations between metabolites, with network module indicated by color ($n = 144$ samples). **b** Metabolites enriched in wet or nonwetted samples, as determined through a two-sided nonparametric $t$ test with $10^5$ permutations. **c** Metabolites enriched in samples originating from an individual inoculating location, with statistical methods as in **b**. **d** Correlations between metadata factors (treated as dummy variables where true $= 1$ and false $= 0$) and the percent of metabolites in network modules. Nonsignificant correlations are not shown

inoculating locations, further suggesting that while location (and hence the primary inoculating microbes) may influence community taxonomic diversity, it does not appear to strongly affect metabolic diversity during growth. The abundance of metabolites in module 7 were anticorrelated with all other modules, but specifically with module 2 (corr $= -0.87$, $p < 0.001$). Module 7 is dominated by wet samples at later time points, suggesting that community succession in wet environments may converge to a common metabolic profile.

**Metabolite features can predict sample type**. Random Forest analysis was employed to determine metabolites associated with various sample types. Models classifying whether a sample had been wetted had an average accuracy of 98% (error ratio $= 25$, with expected random error 0.5), and wetted samples were never misclassified as a nonwetted sample in any of the ten model iterations. Models classifying samples based on material were similarly successful, with an average accuracy of 97% (error ratio $= 25$, with expected random error 0.75). Metabolomics models were much less successful at predicting the inoculating location, with a mean success of 72% (error ratio $= 2.36$ with expected random error 0.67). We sought to gain insight into the chemical composition of metabolites that comprise the signatures observed in these models. Feature importance scores were assigned to compounds based on their relative contributions to predicting sample type. For both the wetting- and material-dependent groups, we selected the 100 highest-scoring metabolite features for further examination and identification (Supplementary Fig. 10). Ninety-eight of the 100 top-scoring metabolites that differentiated wetted and nonwetted samples were enriched in wetted samples. None of these compounds were automatically identified by mzCloud, so the metabolites were analyzed via external database searches, and compound classes were designated based on fragmentation spectra. A diverse set of compound families was observed, including compounds likely to be carbohydrates and glycoconjugates, fatty acids, prenol lipids, sterol lipids, polyketides, and glycerolipids, as well as several pyridine derivatives including a form of vitamin B6, indicative of microbial activity and growth and compounds associated with the surface materials.

Metabolites that were highly enriched in wet vs. nonwetted conditions underwent additional manual analysis for confident structural identification. One of these metabolites was identified as Nigragillin ($C_{13}H_{22}N_2O$, accurate mass $= 222.1723$), which is a fungal alkaloid first identified in *Aspergillus niger*[27]. Nigragillin abundance was significantly enriched in wet MDF and OSB samples (505- and 280-fold, respectively) compared to nonwetted samples. However, no significant differences in nigragillin were observed for gypsum or MF-gypsum. In both wet MDF and OSB the nigragillin concentration increases over time (Supplementary Fig. 11). Another high-scoring metabolite showed MS/MS fragmentation consistent with Fumigaclavine C ($C_{23}H_{30}N_2O_2$, accurate mass $= 366.2291$), which is a fungal alkaloid first identified in *Aspergillus fumigatus*[28]. Fumigaclavine C was enriched in wet samples of gypsum, MDF, and OSB (23-, 26-, and 13-fold increase in comparison to nonwetted samples, respectively), with equivalent abundance in mold-free gypsum regardless of wetting. While the concentration of Fumigaclavine C remained flat or increased slowly in most materials, wet gypsum showed a dramatic increase in abundance at 15 and 20 days post incubation (Supplementary Fig. 11).

Metabolites that were predictive of material type (OSB, MDF, Gypsum and MF-Gypsum) were also further analyzed to determine how these materials influence the chemical composition of metabolites. Of these metabolites, 80% eluted with a retention time of >7 min, indicating a skew toward more hydrophobic compounds. This suggests that hydrophobic compounds are more diverse between the materials and therefore could have greater influence on microbial metabolism than the ubiquitous hydrophilic components. Two of these metabolites were identified by MzCloud search: glucose-phosphate, which was about tenfold less abundant in MF-gypsum compared to all other materials, and scopoletin, a metabolite produced by plants

that has antimicrobial activity[29–31], which was about 60-fold more abundant in MDF samples than in other materials and could be influencing the reduced bacterial growth on this material (Supplementary Fig. 11). Thiabendazole and azoxystrobin, known antifungal compounds[32,33], were highly overrepresented on MF-Gypsum, 333- and 595-fold respectively more abundant than the average content for the other three materials, and as such are likely some of the active compounds in MF-Gypsum.

**Microbe−metabolite co-occurrences.** The abundances of Nigragillin and Fumigaclavine C were each significantly positively correlated with a fungal OTU annotated to the phylum Ascomycota (corr = 0.66, FDR $p = 0.0004$), which contains species known to produce these two alkaloids. Both Nigragillin and Fumigaclavine C have been reported to display antibacterial activity[34,35]. Interestingly, Nigragillin was negatively correlated with the abundance of *Bacillus* and *Pseudomonas* OTUs; this could suggest fungal competition for space and resources[36] against bacteria, and in the specific case of MDF, when Nigragillin abundance was greatest no bacterial growth was detected (Supplementary Figs. 2 and 11). The abundance of glucose-phosphate was significantly correlated to the proportion of a dominant Enterobacteriaceae OTU, a genus which is known to synthesize it[37] (corr = 0.72, FDR $p = 0.000002$). Thiabendazole was positively correlated with *Penicillium* abundance (corr = 0.80, FDR $p < 10^{-9}$). As thiabendazole is prevalent and persistent in the natural environment, this correlation may indicate the presence of thiabendazole-resistant *Penicillium* strains colonizing the material from the built environment[38].

Co-occurrence networks were constructed between the bacterial OTUs and metabolites (SparCC correlation of >0.4; Supplementary Fig. 12) to explore further specific microbe−metabolite associations and possible mechanistic interactions. Significant correlations were observed between *Bacillus* OTUs and a number of different lipid classes which have been previously implicated in either formation or disruption of biofilms[39–42]. In addition, azoxystrobin as well as several lipids were positively correlated with *Bacillus* and negatively correlated with *Pseudomonas*, whereas scopoletin was positively correlated with *Pseudomonas* and negatively correlated with *Bacillus* (Supplementary Fig. 13). These additional antagonistic compound interactions between *Bacillus* and *Pseudomonas* could represent either competitive interactions between these organisms or different adaptation to the different materials and wetting conditions.

## Discussion

As expected, wetted materials had higher bacterial and fungal growth rates and were dominated by a few particular microbes, most notably the bacterial genera *Bacillus*, *Erwinia*, and *Pseudomonas* and the fungal genera *Eurotium* and *Penicillium*. This dominance led to an overall lower alpha diversity compared to nonwetted coupons. Wetting condition and material type described the majority of the variance in bacterial, fungal and metabolite structure. Interestingly, each wetted material showed its own unique microbe−metabolite dynamics.

Gypsum and MF-gypsum were mostly colonized by *Bacillus*, with gypsum being a less selective environment, which allowed for several bacterial species to thrive on the same coupon simultaneously, each of them with high relative abundance and apparently sharing both the physical space and resources. In contrast, MF-gypsum prevented most fungal growth and allowed *Bacillus* to dominate with little competition. MDF selected for fungal growth primarily, which allowed for the rapid accumulation of the antibacterial chemical, nigragillin, which is known to be made by the *Aspergillus* fungi. On OSB material, nigragillin

and fumigaclavine C, a second fungal-synthesized antibacterial metabolite, may play important roles in microbial growth dynamics. Nigragillin, Fumigaclavine C, and *Aspergillus* relative abundance each gradually increases over time, whereas the abundance of *Pseudomonas* declines after the antibacterial metabolites reach peak abundance (Supplementary Fig. 11). These observations bolster our hypothesis that production of antibacterial metabolites by *Aspergillus* may inhibit the proliferation of surrounding bacteria. Also, there is a human health risk associated with the proliferation of the *Aspergillus* fungi in the BE. While the most common species identified in our data was *Aspergillus penicillioides*, a common indoor fungus in damp buildings with known associations to allergies and asthma[43,44], other *Aspergillus* species are known to be able to produce mycotoxins (including aflatoxins), molecules that have been associated with cancer and immunosuppression on humans[17]. Of course, our correlation-based analyses do not definitively establish interactions between taxa or the origins of individual metabolites. Still, we believe these insights will be useful in generating testable hypotheses for future, more specifically targeted studies.

Traditional wood-based building materials contain natural polymers such as cellulose and lignin that are susceptible to degradation by fungal colonization[19,21]. With some fungi having been shown to produce mycotoxins including aflatoxins that could affect human health[17,45], building materials such as mold-resistant gypsum have been developed, which contain antifungal compounds intended to discourage fungal growth. We were particularly interested to examine the microbial communities on these surfaces and as expected, found that fungal growth was diminished on MF-gypsum compared to other materials. However, it appeared that the scarcity of fungal colonies made way for bacterial species to flourish with less competition; on nonwetted materials we observed MF-gypsum bacterial particle counts greater than on the other three materials, and on wetted materials while the MF-gypsum bacterial counts were second to MDF, the abundance level between nonwetted and wetted coupons, unlike MDF, remained minimally changed. This raises the potential that pathogenic bacteria colonization could occur on MF-gypsum and if wetted could grow and lead to negative health outcomes. In terms of metabolite production, thiabendazole and azoxystrobin were some of the antifungal compounds found in high abundance and overall a similar subset of compounds accounted for most of the metabolite abundance on this material, indicating lower metabolic diversity when the colonization is dominated by bacterial growth. We also detected a correlation between thiabendazole and *Penicillium*, which suggested the persistence of thiabendazole-resistant fungal strains.

Additionally, certain lipid metabolites (indicative of biofilm formation) showed significant positive correlation with both *Bacillus* and *Pseudomonas* OTUs, and these lipids were negatively correlated with the abundance of chloroplast OTUs, indicating that when bacteria and metabolites indicative of biofilm formation are detected in greater abundance, we see a proportional reduction in plant-associated signal. Similar to the lipids, metabolites annotated to organic molecules and vitamins were also negatively correlated with the chloroplast OTUs, which suggests that bacterial growth, indicated by increased proportion of 16S, cellular counts and associated metabolites, tends to swamp out the background material-chloroplast signal. We hypothesize that this may be because these molecules are being produced by bacteria colonizing and forming biofilms on the woody material. When the relative abundance of the bacteria increases, it reduces our ability to detect chloroplast sequences (based on a given sequencing depth); as such this negative correlation is likely due to the increased abundance of the microbes that mediate the production of these metabolites, reducing the detection frequency

of specific chloroplast OTUs, and not due to some mechanistic relationship between the wood and these molecules.

Pseudomonads and Bacillus are often the main contributors to biofilm formation on material surfaces in the built environment[46–48]. Biofilms are complex extracellular matrices formed by bacteria through the excretion of lipopeptide biosurfactants, to provide attachment to a surface to support colocalization with a nutrient source and protection from dehydration and chemical activity. Some of these lipopeptide biosurfactants produced by Pseudomonas and Bacillus species have been shown to have lytic or growth-inhibitory activity against many microorganisms such as bacteria, viruses, mycoplasmas, and fungi[48]. Powers et al.[47] demonstrated that Pseudomonas protegens produces antibiotics that inhibit biofilm formation and sporulation in Bacillus subtilis. They also found that Pseudomonas putida secretes an unknown inhibitory compound that prevented biofilm-associated gene expression. In our study we demonstrate a number of compounds known to have potential biofilm inhibitory qualities that also co-correlate with either Pseudomonas or Bacillus abundance, suggesting potential competitive activity between these organisms. While Pseudomonas–Bacillus interactions have been shown to be competitive, interspecies interactions within the genus Bacillus are also important in the formation of biofilms, lipids like hydroxy fatty acids and mono-acyl-glycerophosphocholines could be building blocks or residual products of the biofilm creation[39,40,49].

The simultaneous collection of environmental, metabolomic and microbial profiles reveals insights into the chemical signals that may govern BE microbial communities under high humidity conditions, and provides evidence that these taxa compete for space and resources. Here we show that wetting condition can profoundly alter both fungal and bacterial community succession, and that the taxa which dominate samples after wetting or exposure to high humidity are not abundant in nonwetted materials and have little relation to the skin-associated taxa which dominate samples of indoor environments. After wetting, the microbial community undergoes a successional trajectory that can result in convergence of metabolic diversity even when taxonomic diversity remains variable. We further show that while material choice significantly influences bacterial diversity, the same is not true of fungal diversity. In summary, BE microbial ecology once seen as a wasteland[50] could rather be seen as a desert environment mostly formed with smaller assemblages that can rapidly become an active ecologically dynamic community if water, in liquid or vapor form, is added. When a material experiences high moisture conditions, both fungal and bacterial growth rapidly accelerate and the metabolites associated with their adaptation to different surface materials and competition for resources demonstrate ready-made eco-evolutionary adaptation to this sporadic availability of a crucial resource; this phenomenon is very similar to what has been observed in real desert soil microbiomes[51], as well as in very different ecosystems, such as sediments exposed to oil pollution[52].

## Methods

**Test materials**. Four building materials were used in this study: OSB, MDF, regular gypsum wallboard, and mold-resistant (i.e., mold-free, or "MF") gypsum wallboard. All samples were purchased new from a home improvement store in Chicago, IL. The building materials were cut into 5 cm × 5 cm coupons for testing. The coupons were sterilized by UV irradiation for 20 min followed by surface cleaning with a 70% ethanol solution. This approach likely did not render the coupons DNA free but certainly nonviable.

**Inoculation**. The material coupons were naturally inoculated by passive settling for about 50 days each at one of three locations: two private residences (Location 1 and Location 2) and in a laboratory where they were subsequently incubated (control). The control coupons were kept covered by aluminum foil and kept in the

laboratory for the same duration to minimize natural inoculation by deposition. These control samples were initially treated as a unique location in our analyses and can be thought of as lab-inoculated (albeit with minimal environmental influence) rather than residence-inoculated. Each set of test coupons included 44 coupons for each type of building material (i.e., 176 coupons in total) to allow for multiple subsequent sampling strategies. One set of test coupons was placed inside a sixth floor apartment unit with two adult occupants and a medium-sized dog located in downtown Chicago, IL (Location 1). The other set of materials was placed inside a two-story single-family residence without any pets near the main campus of Illinois Institute of Technology, approximately 8 km south of the downtown residence (Location 2). During the inoculation periods, built environment metadata[53] were collected in each residence, including temperature (T) and relative humidity (RH) using Onset HOBO U12 data loggers and occupant presence within the immediate vicinity of the samples using Onset UX90 data loggers. The UX90 occupancy sensors were placed on the floor next to the samples, facing up, to record movements within the sensor's field of vision as an indicator of how often occupants were in close proximity to the surfaces, which may have affected natural inoculation through direct human shedding. Coupons at a third location (the Built Environment Research Laboratory at the Illinois Institute of Technology) were covered with aluminum foil to minimize natural inoculation, serving as a control group.

**Wetting and incubation**. After inoculation, half of each set of materials (i.e., 22 coupons each) from each location, as well as 22 coupons from the control group, were submerged in tap water in separate pans for ~12 h to simulate the process of wetting of building materials by potable water. The other half of each set of materials (i.e., the other 22 coupons each) from each location and the other 22 coupons from the control group were not submerged in water. Next, just about 10 min after the submersion, to encourage fungal growth on all of the building materials, all of the coupons were placed in trays (each tray contained all 22 coupons of one type of material from one location or control group) and were incubated at room temperature (24 ± 2.7 °C) inside a static airtight chamber (0.9 m × 1.2 m × 0.4 m). Salt solutions (potassium nitrate) were used to maintain constant high RH near ~94% for the duration of the experiment. Constant high RH may not accurately reflect realistic building conditions but was used to encourage growth and has been used in many prior investigates[7,20]. Temperature and RH in the chamber were also recorded using Onset HOBO U12 data loggers.

**Sampling procedures**. The building material coupons were sampled for offline biological and chemical analysis every 5 days at seven different sampling time points. The initial samples (day 0) were taken just after retrieving the samples from the field inoculation and represent nonwetted, naturally inoculated samples previously held at normal indoor environmental conditions. The remaining sampling time points occurred every ~5 days. At each time point, a new coupon of each material from each condition that had never been swabbed before was swabbed, while duplicates of previously unswabbed samples were also swabbed periodically (at 0, 10, 20, and 30 days) for comparison. Two samples were also swabbed at every time point to investigate whether repeatedly swabbing the surfaces impacted the results. Duplicates of both previously swabbed and previously unswabbed samples were also included to investigate whether or not natural inoculation and subsequent growth was evenly distributed across multiple coupons. Supplementary Fig. 1 illustrates the experimental setup and Supplementary Fig. 14 shows coupons' photographs at day 25 and day 30 for each one of the three locations. Details of swabbing procedure at each time point are described below.

First, sampling reagents were prepared as follows. Phosphate-buffered saline (PBS) was used for microbial samples that were to be analyzed for DNA and formaldehyde was used for microbial samples that were to be analyzed by microscopy. For PBS, 500 μL 1× PBS was added to 1.7 mL microtubes for each sample to be collected. For formaldehyde, 100 μL 4% paraformaldehyde was added to 1.7 mL microtubes for each sample. Microcentrifuge tubes were filled with ethanol solution (200 μL 50% EtOH) to preserve samples for surface chemistry/metabolomics analysis.

For subsequent DNA sequencing and analysis, the entire surface of the test coupons was swabbed using two BD Screw Cap SWUBE™ Polyester swabs for 20 s. The same researcher swabbed every time, intentionally applying approximately the same pressure and swabbing in the same pattern to keep the swabbing process consistent. Although polyester swabs have been shown to be less efficient in recovering microbial biomass from surfaces than some other materials such as nylon[54], they were used in a consistent manner and have been shown to perform well vs. roller samplers and electrostatic wipes when controlling for the actual area sampled[55]. Swabs also allowed for minimal disturbance on our small surface area coupons compared to other sampling approaches. One of the double swabs was placed into the tube with PBS and frozen for shipping for subsequent sequencing. The tip of the other of the double swabs was placed into microtubes and the swab tips were vortexed for 10 s. One hundred microliters of sample buffer was removed added to the tube containing 100 μL 4% paraformaldehyde for fixation. These fixed samples were stored in a refrigerator at 4 °C and then sent to the San Diego State University team for running numerical counts of cells and virus particles using microscopy. Negative controls (swabs taken immediately after sterilization) were not included in this study.

For surface and microbial chemistry analysis (i.e., metabolomics), another test coupon was swabbed using a cotton-tipped applicator that is dipped in ethanol[56]. The end of the swabs were cut directly into pre-prepared collection tubes, stored at 4 °C for 2–3 h, and then stored at −20 °C overnight. Swabs were then removed with clean forceps the next morning, then re-sealed into the microcentrifuge tubes and sent to the Northwestern University team on ice at −20 °C or lower. Overhead photos of each tray of coupons were also taken at each time point for image analysis using ImageJ to calculate the percentage of visible microbial growth coverage[20].

**Viral-like particle and bacterial microscopy counts**. Epifluorescence microscopy was used to ensure that all samples contained bacteria and virus-like particles and to estimate their abundance. One hundred microliters of the paraformaldehyde-fixed samples was resuspended into 5 mL of sterile 0.02 μm filtered water. Each suspended sample was then filtered onto a 0.02 μm Whatman Anodisc filter membrane[57]. The filters were stained with 1× SYBR Gold and incubated for 10 min in the dark. Each filter was washed and mounted onto slides to be observed. Visualization was performed using a QImaging Retiga EXi Fast Cooled Mono 12-bit microscope and Image-Pro Plus software was used to collect digital images and estimate VLP and bacterial abundances. We did not estimate fungal cell counts because of the difficulty in counting multicellular hyphae.

**Metabolomics analysis**. Samples were analyzed by high-performance liquid chromatography and high-resolution mass spectrometry and tandem mass spectrometry (HPLC-MS/MS). Specifically, the system consisted of a Thermo Q-Exactive in line with an electrospray source and an Agilent 1200 series HPLC stack including a binary pump, degasser, and autosampler, outfitted with a column (Waters XBridge BEH Shield RP18, 100 × 2.1 mm, 5 μm particle size with matching guard). The mobile phase A was $H_2O$ with 0.1% formic acid; B was acetonitrile with 0.1% formic acid. The gradient was as follows: 0–0.5 min, 98% A; 5 min, 80% A; 10–10.5 min, 5% A; 10.6–15 min, 98% A, with a flow rate of 400 μL/min. The capillary of the ESI source was set to 275 °C, with sheath gas at 40 arbitrary units and the spray voltage at 4.0 kV. In positive polarity mode, MS1 data were collected at a resolution of 35,000. The precursor ions were subsequently fragmented using the higher energy collisional dissociation (HCD) cell set to 30% normalized collision energy in MS2 at a resolution power of 17,500. Data were processed with Compound Discoverer 2.0 (Thermo Fisher) with MS/MS metabolite identifications made by comparing experimental MS/MS spectra with library spectra from MZCloud (lower cutoff score of 90% match).

For the metabolites that were selected for more in-depth characterization, classification of structure or substructure was performed by searching databases such as the Dictionary of Natural Products, the LIPID MAPS Structure Database, and GNPS (Global Natural Products Social Molecular Networking). Predicted structures resulting from a matched intact mass (≤10 ppm error) were subsequently validated through manual analysis of fragmentation mass spectra.

Metabolite differential abundances (fold calculations) were calculated from Compound Discoverer median peak areas for each compound including all three sampled locations.

**DNA extraction and sequencing**. To perform DNA extraction, the Qiagen DNeasy Powersoil HTP kit was used with a modified protocol optimized for low-biomass samples. Swab tips were inserted into each well of the bead plate, and then cut off using a sterilized wire cutter. The manufacturer's protocol was then followed, with the following modifications: before cell lysis, the bead plates (containing beads, bead solution, swabs, and the C1 solution) were heated for 20 min at 60 °C in a water bath. Additionally, the protocol steps using solutions C2 and C3 were combined into a single step, by adding 150 μL each of C2 and C3 together to the lysed sample in the 1 mL plate.

The DNA obtained from the DNA extraction was used for both high-throughput 16S/ITS sequencing, and qPCR. The 16S sequencing targeted the V4 region of the bacterial 16S rRNA gene, using the primer pairs 515F/806R. The ITS sequencing targeted the highly variable fungal internal transcribed spacer region 1 located between the 5.8S and 18S rRNA genes, using the ITS1f and ITS2 primer pairs[58]. Both primer sets used the same reaction mix and thermocycler instructions: Reaction mix: 9.5 μL of molecular biology grade $H_2O$, 12.5 μL of Accustart II PCR Toughmix, 1 μL each of forward and reverse primers at 5 μM, and 1 μL of sample DNA for a total reaction volume of 25 μL.

To make both the 16S and ITS amplicons, the following PCR program was used: Initial denaturing step at 94 °C for 3 min, followed by 35 cycles of: 94 °C for 45 s, 50 °C for 60 s, and 72 °C for 90 s, followed by a final extension step of 72 °C for 10 min. The resulting amplicons were quantified using the Picogreen dsDNA binding fluorescent dye on a Tecan Infinity M200 Pro plate reader and pooled to 70 ng DNA per sample using the Eppendorf epMotion 5075 liquid handling robot. Primers and PCR reagents were removed using Agencourt AMPure beads, and then the clean amplicon pool was sequenced at Argonne National Laboratory's Environmental Sample Preparation and Sequencing Facility, following the Earth Microbiome Protocol[59]. Sequencing was performed on an Illumina Miseq using V3 chemistry, generating 2 × 150 nt reads. For the fungal sequencing, 2 × 250 nt reads were generated by using additional cycles.

In addition to our BLP counts, we employed qPCR to estimate bacterial abundance. We view these approaches as complementary as they are subject to different sources of bias. In microscopy, error comes from swabbing and preparation, while in qPCR, error can originate from DNA extraction, the PCR reaction, PCR primers not binding equally to all sequences, and 16S copy number. qPCR was performed using a Roche LightCycler 480 II. The 515F/806R primer pair was used again for amplification, using a mix of 10 μL LightCycler 480 SYBR Green I Master mix, 6 μL of molecular biology grade $H_2O$, 1 μL of 515F primer (10 μM), 1 μL of 806R primer (10 μM), and 2 μL of template DNA for a total of 20 μL per reaction. The following thermocycler conditions were used: (1) 95 °C for 5 min, (2) 95 °C for 10 s, (3) 45 °C for 45 s, (4) Measure fluorescence, with steps 2 through 4 repeated 50 times. To determine the copy number of the 16S gene (and therefore the number of organisms per swab), a standard curve was generated using a serial dilution of a plasmid containing the *Escherichia coli* 16S rRNA gene.

For fungal qPCR, each 20 μl reaction contained 1 μL each of the forward ITS1f (CTTGGTCATTTAGAGGAAGTAA) and reverse ITS2 (GCTGCGTTCTTCATC GATGC) primers at 10 μM, 10μl of the mastermix, 6 μl of DNA-free water, and 2 μl of the target DNA extraction. The target was amplified using the following conditions: Initial denaturing step for 15 min at 95 °C, following by 45 cycles of a 95 °C denaturing step for 1 min, an annealing step at 53 °C for 30 s, and an extension step of 72 °C for 1 min. The amount of amplified DNA was quantified after each extension step. A standard curve was constructed using the Zymo Femto Kit, which contains serially diluted DNA from *Saccharomyces cerevisiae* TMY18. Using this curve, the copy number per μl for each extraction was determined.

Forward reads were quality-trimmed and processed for OTU clustering using the open reference method implemented in the QIIME pipeline[60]. The sequence identity cutoff was set at 97%, and taxonomy was assigned to the high-quality (<1% incorrect bases) candidate OTUs using the parallel_assign_taxonomy_rdp.py script of the QIIME software.

**Treatment of technical replicates**. We used Mantel test to determine whether the bacterial communities on replicate coupons (coupons on the same tray sampled at the same time) significantly resembled each other and preserved patterns of beta diversity. We began by calculating the Bray−Curtis dissimilarity between each pair of samples taken from the same material type using the *beta_diversity.py* function from the software QIIME 1.9.1 [60], producing dissimilarity matrices for each sampling type. Then the Mantel test and false discovery rate adjustment was performed using the *mantel* and *p.adjust* functions in the Vegan and stats R packages. For all comparisons between the sampling types, the Mantel statistic (which measures the stress in the fit of the two matrices) was significantly high (Mantel ≥0.67 for fungi and ≥0.5 bacteria (all $p < 1E-05$)) (Supplementary Tables 2 (Fungi) and 3 (Bacteria)). Based on the highly significant resemblance between material types, we treated all samples of the same type as technical replicates, meaning that all combinations of material, location, wetting condition, and time point had either 2 (time point 0), 3 (time points 1, 3, and 5), or 4 (time points 2, 4, and 6) replicates.

**Rarefaction and statistical analyses**. After sequencing and sample merging, bacterial and fungi OTU tables were rarefed to 1000 and 10,000 reads respectively for statistical analyses. Rarefaction and statistical analyses were performed using R.

**Random forest analyses**. Random forest models were implemented using the randomForest R package. Samples from time point 0 were removed from the dataset. Models were built with 1000 trees and tenfold cross validation. For each of the ten models for each metadata criterion, a randomly drawn 70% of samples (100 samples) were used for model training and the remaining 30% (44 samples) were used for validation.

**NMDS**. We visualized sample similarity using NMDS ordination based on Bray−Curtis dissimilarity. Metadata vectors were fit onto the ordination using the *envfit* command in the Vegan R package. We converted material, location, and wetting condition into dummy variables (1 = yes, 0 = no) and, in the case of the bacterial and fungal datasets; also fit vectors of relative abundance for the common genera described in Fig. 3. We assessed significance of each of the vectors using $10^5$ permutations, and removed nonsignificant vectors from the figure. The $R^2$ values for each vector and their significance are presented in Supplementary Table 1.

**Co-occurrence networks**. Traditional correlation networks are unsuited to genomic survey data as these data are relative, rather than absolute, measures of community composition. Since the relative abundances of all taxa within each sample must sum to 1, the fractions are not independent and will often exhibit negative correlations to each other regardless of the true correlation in absolute abundance. To avoid these compositional effects, we generated our networks using SparCC[26], a correlation metric based on log-ratio transformed data that is specifically suited to compositional genomic surveys. Pseudo-$p$ values for each correlation were generated through comparison from 100 to 1000 bootstraps of the permuted OTU table.

For the same kingdom and microbe−metabolite networks only samples where either bacteria or fungi and metabolites were found in detectable levels after

rarefaction were used ($N = 83$, $N = 91$ respectively). Additionally, only bacterial OTUs with >9 reads, fungi OTUs with >99 reads, and metabolites with >5,000,000 abundance in the rarified dataset were used for a total number of 630 bacterial OTUs, 352 fungi OTUs and 426 metabolites. Figures were generated using CAVNet R package[61] and only displayed the higher correlation threshold (positive or negative) greater than 0.4.

For the network encompassing both bacteria and fungi, the OTUs reads threshold remained the same but only samples with both 16S and ITS data were kept ($N = 153$) producing a new subset of bacterial and fungi OTUs, 590 and 581 OTUs respectively. This dataset produced a co-occurrence network with 1171 nodes. Only positive correlations with a pseudo-$p < 0.05$ were included, resulting in a network with 33,509 edges (density = 0.052). The network was ordinated using the Fruchterman−Reingold Algorithm (edge-weighted, force-directed) in the *igraph* R package, with node size based on the log read count of each OTU across all samples (with ITS counts first divided by 10 to equalize rarefaction depth between datasets). We used the Walktrap method[62] to uncover dense subgraphs (modules) within the network, which may correspond to distinct community structures. We chose Walktrap (which is based on random walks within the network) as our method of community inference due to its computational tractability and its accuracy at uncovering subgraphs regardless of network size[63]. We used random walks of four steps, which resulted in four distinct modules with a network modularity of 0.45.

**Reporting summary**. Further information on research design is available in the Nature Research Reporting Summary linked to this article.

## Data availability
The raw data from this project have been loaded to the QIITA database[64] with study number 11950 as well as the European Nucleotid Archive from EMBL-EBI with study code PRJEB29658. OTU tables for the 16S and ITS datasets, the metabomomics dataset, the VLP and BLP counts, and the associated mapping files via FigShare at https://doi.org/10.6084/m9.figshare.7865015.v2.

## Code availability
The code used for the analyses will be made available upon e-mail request to the authors.

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

## Acknowledgements

The authors wish to thank the Sloan Foundation for financial support of the study.

## Author contributions

J.A.G., B.S., S.T.K. and P.M.T. conceived of the study. S.L., C.C., D.Z., V.J.W., G.G., P.G., N.G., E.M.H. and C.H. collected and analyzed data. S.L., C.C., V.J.W., B.S. and J.A.G. wrote the paper. All authors edited and approved the paper.

## Additional information

**Competing interests:** The authors declare no competing interests.

