## [Peer Review File · Nature Communications]

Reviewers' Comments:

Reviewer #1:

Remarks to the Author:

This paper describes an extensive experimental system to study microbial and metabolic signatures on various building materials incubated at high humidity. The tools used were broad (particle counts, qPCR, amplicon sequencing, metabolomics) and the experimental design extensive. That the identity of so many of the chemicals had to be manually investigated shows how much we have to learn about the chemicals inside our buildings.

I see several aspects of the paper that currently limit its novelty and impact. First, ecological interactions of succession, competition, interference, etc, are inferred but not demonstrated. While the conclusions the authors draw are logical scenarios, they remain hypotheses. The language and implications should reflect that the conclusions represent ideas in need of further validation. In one prominent example of this, the chemical compounds are referred to as microbial metabolites; the authors themselves acknowledge that many of these compounds are likely coming from the building material itself. Also, many of the observed compounds could simply be tracers of microbial growth, and any effect on ecological dynamics is speculative.

With regards to data analysis presentation, there are several gaps in detailing the study protocol. How were the sequences processed in order to infer OTUs? Specifically, for fungi, it's not mentioned which ITS region was targeted (specifying the primers would aid in that), nor how the 2x150bp sequences were handled. The ITS1 and ITS2 regions range in length, but with 2x150bp, many amplicons would not overlap, and how this was handled was not presented. Undoubtedly this short sequence length drove the low taxonomic assignments in the dataset. Also, the compositional nature of the amplicon data was used to justify generating correlation networks with SparCC, but community analysis (distance and ordination) was not handled in a compositional way (perhaps drawing on Atchison and PCA rather than using Bray-Curtis and NMDS). How long after collection from the various locations were half the samples submerged in water? Likewise, how long after being submerged were the samples put in incubators? The qPCR protocol for fungi was not included. Further explanation on why the authors think repeated sampling does not affect communities would also be helpful. Although the outcomes are highly statistically correlated, the strength of the mantel correlation is ~ 0.6 (Table S3, not Figure S3 as is mentioned in the table). Why is it appropriate to use mantel tests here but not for correlation networks?

Points of clarity:

- Figure S1 does not explain why it is appropriate to use all samples of the same type as technical replicates.
- L107: There is no explanation what control and non-control samples refer to; it was eventually inferred that "control" was linked to Location and meant the samples that remained in the lab, while non-controls referred to those samples placed in buildings for natural inoculation. Given the eventual finding that the "control" samples are not, in fact, sterile, control is not a clear way of explaining this samples. Perhaps the inoculation potentials of the samples are better described as "Sealed in lab", "House 1" and "House 2." Along the same lines, Control/Location in Figure 1 is vague. Moreover, it is not sufficiently explained how the authors decided that the control samples were similar enough to the house-inoculated samples to lump them together, particularly when they go on to argue that Location had a meaningful, although weak, effect on the resulting bacterial and fungal communities.
- While I appreciate that the point of the Experimental Setup paragraph is to briefly summarize the methods before describing the results, the level of detail and information is inconsistent. For example, precise values of the occupancy values and rarefaction levels were included, but major aspects of the study execution were left to the Materials and Methods. For the occupancy levels, the values of $< 0.2\%$

seem very low (and it's hard to imagine that this percentage of occupancy time would drive any differences in overall exposure), but the context for these values is never discussed. Is that typical for homes? These kind of details seem more appropriate for the Materials and Methods.

- The authors go back and forth between referencing 16S/ITS and bacteria/fungi. Is there a strategy behind this or a consistent pattern to the notation? Similarly, in the section "Visible growth, particulate counts and qPCR" and Figure 1, the authors inconsistently refer to BLP and Bacteria, when it seems like it should always be BLP when discussing the results of that method.
- L244: What is the biological significance of "increase in the absolute number of significant co-occurrence relationships" and the implications of the number of edges?
- L417: similar metabolic diversity - similar to what?
- L475: Comparing the BE to a desert has been argued before, including in the literature by Gibbons, 2016. It seems to me that the data here are consistent with that article.
- In Figure S11, it would be slightly easier to read if the pieces of the legend were switched.
- In the Introduction, the authors write, starting Line 81: Improved understanding of how bacterial and fungal metabolism is shaped by environmental properties (e.g., the presence of water, surface material composition) and inoculating source (e.g., building location, occupancy patterns) could have important implications for architectural design, construction, building management, and occupant health (Rand et al., 2017). Given the study findings, are there any implications for design, construction, management, and occupant health? This conjecture is absent from the discussion, and I suggest either adding it, or amending this part of the introduction.
- In my opinion, more attention needs to be given to the wetting process and what that means for the study, particularly in regard to the inoculation location. The study is, in essence, asking how a one-time pulse of water stimulates a community that then experiences conditions favorable to growth, in comparison to a community in conditions favorable to growth without that previous pulse event. One way to interpret the results is the simple scenario that the flooding event stimulated the growth of *Bacillus* and *Penicillium* from those samples in the lab and Location 1, while *Pseudomonas*, *Erwinia*, and to a lesser extent, *Penicillium* were stimulated from those samples in Location 2. These stimulated microbes were then able to persist or grow under elevated humidity, in contrast to non-wetted samples where only those microbes able to proliferate under elevated humidity alone (*Eurotium*) were observed. The observed chemicals on these resulting surfaces could then be interpreted in light of these growth scenarios and conjectures made. How did the bacterial biomass as determined through 16S qPCR in non-wetted samples change over time, in comparison to the wetted samples? (The reported results seem to include both non-wetted and wetted samples together). It would be interesting to know qPCR results at T0 for each of the conditions to inform the implications of the one-time pulse event.

Grammatical Typos

- L21: Do not need comma after Nigragillin
- L104: to "the" same sequencing depth
- L108: should say looked very similar to non-control samples?
- L111-112 "Highlighting..." Sentence fragment
- L126 extra close parentheses
- L235: Incomplete sentence
- L333: odd wording "a metabolite produced by the plants"
- L334: need parenthesis: with antimicrobial activity (Lerat et al., 2009...
- L438: missing period.
- L440: MICROBIAL-METABOLITE INTERACTIONS. Do not need the period at the end. (Maybe that's where the period from L438 ended up)
- L449: Powers et al (YEAR)

Reviewer #2:
Remarks to the Author:
GENERAL RESPONSE

I. CONTEXT and ORIGINALITY

The scientific this manuscript —Characterizing microbial activity on building materials— is a contemporary and (re)emerging topic in the industrial hygiene arena.

There is clearly a need for more studies to evaluate the induction, progression, “colonization”, metabolomics and associated (bio)deterioration of building materials. In this reviewer’s opinion, this has the potential for relatively broad interest, but more from a public health standpoint than microbial “community/metabolism dynamics”. I am not sure the salient theme of this manuscript with is consistent with the mission of this particular version of the Nature Journal series; is NC is the best place to introduce this level of multi-dimensional material survey (this is an editorial call). There are specialty journals in this sub-discipline (e.g. Journal of Biodegradation and Biodeterioration, Built Environment, etc.)

There are many authors on this manuscript; most this reviewer recognizes for specialty skills related analyses of environmental samples (e.g. microscopy, MS, bioinformatics) from the Built Environment (BE). This reviewer recognizes the “greater” design of this study is noble, clever and warranted: time-series of samples are recovered from a relatively large cohort of building material coupons (with actual and pseudo-replicates), which are “aliquoted”, split and analyzed for different biomarkers (biochemistry, PCR, genomics) in parallel. The authors then use advanced statistical analyses to discern correlations and co-occurrences of biomarkers, with different resolution. A tricky task to say the least, when managing different sampling and analytical needs that must “tie-together” otherwise very disparate skill sets. Therein lies the challenge and value of this work.

This reviewer found this work to be largely observational, leveraging statistical outcomes, through a broad range of significance levels, to implicate some generalized conditions and support conclusion. Some of these conditional conclusions appeared tangential and detracted from the core findings (mission) of the study (listed below).

Organization and Readability

This reviewer found this manuscript to be well written; it followed a logical path with coherent organization; however, it was voluminous and tedious to get through (considering the supplementary materials). In terms of readability, the relatively large cohort of statistical tests applied, with the range of significance thresholds, made it difficult to follow at times. I am not sure how to suggest to condense this, except to eliminate some of the lesser important “suggestions” that had no direct biochemical/genomic data to support the stated implications as follows:

Page 9, In 362-365 biofilm formation (biofilm/carbohydrates not measured);

Page 9, line 345, suggestion of generalized alkaloid production by the Ascomycota (alkaloid pools not measured);

Page 9, indication of the presence of R-azole-resistant *Penicillium* species (this ARG not measured), etc.)

Page 10 *Pseudomonas* spp. specific dose-dependent response to antibacterial metabolites (S11).

If the manuscript contents were to be condensed to the context of its title, this reviewer would recommend completely dropping the virus observations, as they detract from the core genomics and metabolite findings (the salient theme of the work). While the influence of viruses "could" be important, there was no data acquired which implicate how viruses may be associated with the microbial community dynamics and selected metabolomics observed.....just VLP counts. This reviewer believes the Prussin et al (2015) report was cited out of context (pg. 4, ln 128) since this pertains to VLP observations from PM recovered from filtered aerosol samples, and not the progression of microbial growth on surfaces (very different in BE context).

Along the same theme, the photographs and generic description of generalized microbial growth ("visible" surface coverage) on the different dry/wet material types does not add scientific value to this manuscript. Visible mold does grow on wetted materials faster than their drier counterparts, and industrial hygienists routinely use visual confirmation of mold growth for building assessments. The photos and accompanying text add sensation and length to the text; but, in this reviewer's opinion, really add nothing toward the core scientific contributions of the manuscript.

While potable water leaks are an appropriate representation for wetting with water source, this reviewer does not believe the water used to wet/saturate the material coupons is in anyway representative of flood-associated waters (abstract and introduction) and their potential as co-inoculums in this context. This reviewer suggests any consideration for this representation (flood waters) should be removed from the manuscript

Specific Technical Concerns

This reviewer presents the following technical concerns in the order of what I consider the most important in terms of supporting the work.

First and foremost is clarity with regard to the quantitative recovery of genetic materials from the different material surfaces, and the ability to reproducibly sequence these materials with high fidelity and minimal bias. What concerns me here is the description of broad spectrum quantitative PCR results as juxtaposed to the direct microscopy results (pg 4, lines 131 through 139). If BLP counts and broad spectrum qPCR were not in reasonable agreement on the different materials, this uncertainty lessens confidence that subsequent sequencing may not accurately represent the community abundance observations on which much of the manuscript is based. While the authors state on pg 4 line 139 that the different methods have different biases, there is no resolution to sequencing uncertainties introduced here (are they systematic biases at least?)

In the context of comparing microscopy to qPCR, the authors present only BLP counts. Given the clear importance of fungi growing in and on wetted building materials, why are fungal spore counts absent from this report? Given VLP and BLP are included, and (much) more difficult to enumerate with direct microscopy than their fungal counterparts, why was the parallel juxtaposition of ITS qPCR and fungal spore counts absent?

Acknowledging that the authors used a defined yet generic swabbing protocol to recover microbial samples, and applied some statistical rubric to define "reproducible" recovery, there is no discussion or acknowledgement regarding efforts to carefully standardize surface sampling. This is not a trivial issue and associated effort has been previously devoted to this topic in recent peer reviewed literature.

Were the UV-ETOH sterilization protocols successful? Generic sterilization refers to a culture-based endpoint. The material coupons were not rendered DNA free (as judged by the analytical protocols, it does not appear so)? If this protocol did not, was the DNA baseline recovered consistent across the respective materials. If the authors could please expand this point and formally qualify sterilization where introduced in the methods section, pg 12, ln 490.

Reviewer #3:

Remarks to the Author:

The manuscript by Lax et al investigates the dynamics of bacterial and fungal communities (and their metabolites) evolving over four surfaces of building materials found (presumably) in indoor environments.

Although the experiment is complex, the generated dataset complete and the analysis extensive, the specific questions and hypotheses leading this study are unclear. The authors had selected four materials, two of which are organic, but this is not fully exploited in the analysis or in the interpretation of the data. In addition, measurements characterizing the effect of biodeterioration on the physical properties of the materials selected are entirely absent. Moreover, the experimental design includes incubation at an extremely high relative humidity over a long period of time, which is not necessarily compatible with environmental conditions in which biodeterioration could occur. The authors should clarify all these points as part of the introduction and justification of the study.

Specific comments:

Line 4-5. Indicating rRNA is misleading as the authors used DNA and not RNA for this analysis. This reviewer initially thought the authors investigated active microbial communities but later understood this was not the case.

Line 71-72. The justification for the choice of the materials, as well as a more thorough description of those should be included.

Lines 91-114. The entire section is very difficult to understand. The section is titled "experimental design" but it is impossible to grasp the meaning of controls, the duration of the experiment, sterilization procedure, analysis performed on the surfaces (other than "looked similar"). All these details are indicated as part of the methods section, but they should be at least indicated briefly if the section is to be maintained. Moreover, the rationale behind the experiment and the hypothesis leading the experiment should be indicated.

Line 199. Please select one nomenclature "coupons" or "tiles" but do not use those simultaneously unless there is different meaning attached to either one.

Lines 502-504. Key environmental information such as spore load at the "inoculation location" should be provided. Also, it is unclear how the data obtained from the sampling sites was integrated in the analysis.

Line 507. Please consider clarifying exactly the handling of the control group. What is the source of microorganisms for the wetted control material? Is it deposition? This is not clear.

Line 522. Please indicate clearly the timing of the sampling points. T0-T6 is not informative and the reader needs to guess the experiment post-wetting lasted around 30 days.

We thank the reviewers for their constructive comments, which we believe have greatly strengthened the revised manuscript. Please find our point by point response to those comments below.

Reviewer #1

This paper describes an extensive experimental system to study microbial and metabolic signatures on various building materials incubated at high humidity. The tools used were broad (particle counts, qPCR, amplicon sequencing, metabolomics) and the experimental design extensive. That the identity of so many of the chemicals had to be manually investigated shows how much we have to learn about the chemicals inside our buildings.

I see several aspects of the paper that currently limit its novelty and impact. First, ecological interactions of succession, competition, interference, etc, are inferred but not demonstrated. While the conclusions the authors draw are logical scenarios, they remain hypotheses. The language and implications should reflect that the conclusions represent ideas in need of further validation. In one prominent example of this, the chemical compounds are referred to as microbial metabolites; the authors themselves acknowledge that many of these compounds are likely coming from the building material itself. Also, many of the observed compounds could simply be tracers of microbial growth, and any effect on ecological dynamics is speculative.

We agree with the reviewer that the observational methods we employ in this study are not capable of establishing specific interactions between different microbial taxa or between taxa and metabolites. We only aim to gain inference from our large co-occurrence dataset that can be used to generate testable hypothesis, which of course would have to be validated in controlled trials. Such experiments would be well beyond the scope of the current paper, but we hope our results will inspire and contextualize future research on the topic. We have edited our language in the discussion section to clarify that the statistical analyses we employ do not definitively establish any relationships between taxonomic groups or metabolites, but rather synthesize general patterns of co-occurrence.

With regards to data analysis presentation, there are several gaps in detailing the study protocol. How were the sequences processed in order to infer OTUs? Specifically, for fungi, it's not mentioned which ITS region was targeted (specifying the primers would aid in that), nor how the 2x150bp sequences were handled. The ITS1 and ITS2 regions range in length, but with 2x150bp, many amplicons would not overlap, and how this was handled was not presented. Undoubtedly this short sequence length drove the low taxonomic assignments in the dataset.

We made several clarifications to the methods section, most notably highlighting the differences of ITS and 16S preparation, sequencing and qPCR, for instance for ITS the read length was 2x250bp and not 2x150bp, using primers ITS1f and ITS2 to target ITS1 region.

Also, the compositional nature of the amplicon data was used to justify generating correlation networks with SparCC, but community analysis (distance and ordination) was not handled in a compositional way (perhaps drawing on Atchison and PCA rather than using Bray-Curtis and NMDS).

We agree that compositional data has broad statistical limitations, but we would argue that these limitations are more pronounced in correlation analyses than in beta-diversity calculations and their resulting ordinations. Our samples, as in most 16S-based community analyses, are normally dominated by a few highly abundant taxa and a long tail of rare taxa. Because of this general abundance distribution, correlations between rare taxa are usually driven by changes in the absolute abundance of the dominant taxa and not by any meaningful ecological interactions, a problem SparCC attempts to rectify. In our beta-diversity calculations, which are always abundance-weighted,

dissimilarly is driven primarily by differences in the relative abundance of the dominant taxa, which is captured by our compositional data. Beta-diversity calculations based on rarefied read counts has long been a key method in microbial ecology, and we believe that our dataset is sufficient to capture the general patterns of community similarity that our NMDS plots are used to visualize.

How long after collection from the various locations were half the samples submerged in water? Likewise, how long after being submerged were the samples put in incubators?

As stated in the manuscript previously, submersion was ~12 hours each (i.e., overnight). We had not stated in the manuscript, however, that the samples were placed in the high RH incubators within approximately 10 minutes after submersion. This has been added.

The qPCR protocol for fungi was not included.

Our mistake, the protocol is included in the methods section now. “Each 20ul reaction contained 1uL each of the forward ITS1f (CTTGGTCAATTTAGAGGAAGTAA) and reverse ITS2...”

Further explanation on why the authors think repeated sampling does not affect communities would also be helpful.

Repeated sampling certainly has some effect on the community but according to our Mantel test results in the section *Treatment of Technical Replicates* from the Methods, there is a strong correlation between the microbial community composition from different sampling types. Granted, Mantel test results < 1 do suggest that repeated sampling and non-repeated sampling do yield meaningful differences. However, neither approach is perfect: repeated sampling is likely to disturb the community on each repeatedly sampled coupon, but non-repeated sampling is likely to result in community dynamics that are different on each coupon and thus may not reliably be sampled over time. With no gold standard and limitations to each approach, we decided to utilize both.

Although the outcomes are highly statistically correlated, the strength of the mantel correlation is ~0.6 (Table S3, not Figure S3 as is mentioned in the table). Why is it appropriate to use mantel tests here but not for correlation networks?

As mentioned above, we believe our compositional data is well suited to capturing general patterns of community similarity (which is what we employ the Mantel test for), but not for establishing interactions between taxa, which are too heavily influenced by the absolute abundance of dominant taxa.

Points of clarity:

- Figure S1 does not explain why it is appropriate to use all samples of the same type as technical replicates.

Figure S1 was meant to visualize the complicated sampling procedure, not necessarily to describe the similarity of technical replicates. However, we have a separate section in the Methods titled *Treatment of Technical Replicates* that addresses that issue. We have corrected the text to refer to Figure S1 only for experimental setup and Tables S2 and S3 for the Mantel test results for sample strategies.

- L107: There is no explanation what control and non-control samples refer to; it was eventually inferred that “control” was linked to Location and meant the samples that remained in the lab, while non-controls referred to

those samples placed in buildings for natural inoculation. Given the eventual finding that the “control” samples are not, in fact, sterile, control is not a clear way of explaining these samples. Perhaps the inoculation potentials of the samples are better described as “Sealed in lab”, “House 1” and “House 2.” Along the same lines, Control/Location in Figure 1 is vague. Moreover, it is not sufficiently explained how the authors decided that the control samples were similar enough to the house-inoculated samples to lump them together, particularly when they go on to argue that Location had a meaningful, although weak, effect on the resulting bacterial and fungal communities.

We agree with the reviewer that it is confusing to refer to a sample group as “control” and then to treat it as a unique location alongside our other two locations. To provide greater clarity, we have emphasized in the manuscript that “control” can be understood as “lab inoculated” rather than “residence inoculated”.

- While I appreciate that the point of the Experimental Setup paragraph is to briefly summarize the methods before describing the results, the level of detail and information is inconsistent. For example, precise values of the occupancy values and rarefaction levels were included, but major aspects of the study execution were left to the Materials and Methods.

We agree with the reviewer that this inconsistent presentation is confusing. We have moved these experimental setup details entirely to the methods section to increase clarity.

For the occupancy levels, the values of <0.2% seem very low (and it’s hard to imagine that this percentage of occupancy time would drive any differences in overall exposure), but the context for these values is never discussed. Is that typical for homes? These kind of details seem more appropriate for the Materials and Methods.

We should clarify that these aren’t really “occupancy levels” but rather are the “percentage of time that the passive infrared occupancy sensors detected movement within the field of vision of the sensor near the samples, which were placed on the floor in a central area, facing up.” So it’s not a measure of building occupancy (which is much higher), but rather “close proximity” periods, as a rough indicator of how often occupants were in close proximity of the surfaces, which was intended to potentially offer inferences about how direct human shedding of microbes may have affected natural inoculation. To our knowledge, there are not robust records of this parameter in the literature to which we can compare. Clarifying text was added.

- The authors go back and forth between referencing 16S/ITS and bacteria/fungi. Is there a strategy behind this or a consistent pattern to the notation? Similarly, in the section “Visible growth, particulate counts and qPCR” and Figure 1, the authors inconsistently refer to BLP and Bacteria, when it seems like it should always be BLP when discussing the results of that method.

We agree with the author that amplified marker gene name (16S/ITS), and the taxa those markers originate from (bacteria/fungi), are used too interchangeably. We have edited the text to use the gene name when discussing molecular methods and node groups in the correlation methods, and “bacteria” or “fungi” elsewhere.

- L244: What is the biological significance of “increase in the absolute number of significant co-occurrence relationships” and the implications of the number of edges?

We believe that the increase on number of co-occurrences suggest that the wetting event has made the environment more suitable for microbial activity and interactions. We have clarified this in the manuscript.

- L417: similar metabolic diversity - similar to what?

We meant to say that wetted samples become metabolically similar to each other. We have edited the text to say “After wetting, the microbial community undergoes a successional trajectory that can result in convergence of metabolic diversity even when taxonomic diversity remains variable.”

- L475: Comparing the BE to a desert has been argued before, including in the literature by Gibbons, 2016. It seems to me that the data here are consistent with that article.

We meant to provide a distinction between a uniformly inhospitable environment (a “wasteland”), and an environment in which conditions can rapidly improve when water is added and microbial life can flourish (a “desert”). Gibbons uses the two terms interchangeably [*“Overall, the authors suggest that BE surfaces are microbial deserts, wastelands like the Atacama Desert, where water and nutrients are scarce”*], and we have made an effort to clarify our distinction in the manuscript.

- In Figure S11, it would be slightly easier to read if the pieces of the legend were switched.

We agree with the reviewer and have altered the figure accordingly.

- In the Introduction, the authors write, starting Line 81: Improved understanding of how bacterial and fungal metabolism is shaped by environmental properties (e.g., the presence of water, surface material composition) and inoculating source (e.g., building location, occupancy patterns) could have important implications for architectural design, construction, building management, and occupant health (Rand et al., 2017). Given the study findings, are there any implications for design, construction, management, and occupant health? This conjecture is absent from the discussion, and I suggest either adding it, or amending this part of the introduction.

Our sense of the practical implications of this work thus far for design, construction, management, or occupant health are not entirely clear at this early stage, or at least not proven. One potential implication results from the fact that material type was not associated with fungal community structure but location, which demonstrates that the fungi that proliferate on materials after experiencing wetting and/or high humidity conditions are more influenced by the natural inocula that settle on the material than by what is inherently embedded in or assembled on a building material. This could potentially mean that one could predict future fungal community structure that results from wetting or high moisture events (and thus infer important information about the potential impact on human health and remediation strategies) by sampling microbial communities prior to the wetting/moisture event occurring (and potentially choosing to alter the community in a preventative effort). However, significant differences in metabolite composition based on material (combined with non-significant differences between locations) suggests potentially otherwise, such that to be able to predict the eventual compounds produced by microbial activity due to wetting/moisture events (which, at least in terms of mVOCs, have some plausible affects on human health), one indeed might need information on material chemical composition. Another potential implication is that because our results show that the nature of wetting (i.e., whether by bulk potable liquid water or just high RH) has important implications for what communities thrive, one could imagine that given two damp buildings, as indicated by having high surface RH upon measurement,

one might want to know whether it was wet because of a leak/flood or because of water vapor migration because it could influence the nature of the community that is growing there and thus what the implications for health might be. Further, interactions between bacteria and fungi remain interesting but we think we have limited predictive ability at this time. Taken together, although we think we have some suggestive implications that this work could have on the identified practical aspects, they are probably not worth a full discussion in the text at this time. Therefore, we have only minimally introduced these ideas in the discussion and have toned down the introduction language regarding practical implications.

- In my opinion, more attention needs to be given to the wetting process and what that means for the study, particularly in regard to the inoculation location. The study is, in essence, asking how a one-time pulse of water stimulates a community that then experiences conditions favorable to growth, in comparison to a community in conditions favorable to growth without that previous pulse event. One way to interpret the results is the simple scenario that the flooding event stimulated the growth of *Bacillus* and *Penicillium* from those samples in the lab and Location 1, while *Pseudomonas*, *Erwinia*, and to a lesser extent, *Penicillium* were stimulated from those samples in Location 2. These stimulated microbes were then able to persist or grow under elevated humidity, in contrast to non-wetted samples where only those microbes able to proliferate under elevated humidity alone (*Eurotium*) were observed. The observed chemicals on these resulting surfaces could then be interpreted in light of these growth scenarios and conjectures made. How did the bacterial biomass as determined through 16S qPCR in non-wetted samples change over time, in comparison to the wetted samples? (The reported results seem to include both non-wetted and wetted samples together). It would be interesting to know qPCR results at T0 for each of the conditions to inform the implications of the one-time pulse event.

This suggestions are useful, we have added the 16S qPCR counts for wetted and non-wetted samples, and also the counts at TPO for both bacteria and fungi.

Grammatical Typos

- L21: Do not need comma after Nigragillin
- L104: to "the" same sequencing depth
- L108: should say looked very similar to non-control samples?
- L111-112 "Highlighting..." Sentence fragment
- L126 extra close parentheses
- L235: Incomplete sentence
- L333: odd wording "a metabolite produced by the plants"
- L334: need parenthesis: with antimicrobial activity (Lerat et al., 2009...
- L438: missing period.
- L440: MICROBIAL-METABOLITE INTERACTIONS. Do not need the period at the end. (Maybe that's where the period from L438 ended up)
- L449: Powers et al (YEAR)

We thank the reviewer for catching these errors. Each has been addressed in the revised manuscript.

Reviewer #2

GENERAL RESPONSE

I. CONTEXT and ORIGINALITY

The scientific this manuscript —Characterizing microbial activity on building materials— is a contemporary and (re)emerging topic in the industrial hygiene arena. There is clearly a need for more studies to evaluate the induction, progression, “colonization”, metabolomics and associated (bio)deterioration of building materials. In this reviewer’s opinion, this has the potential for relatively broad interest, but more from a public health standpoint than microbial “community/metabolism dynamics”. I am not sure the salient theme of this manuscript with is consistent with the mission of this particular version of the Nature Journal series; is NC is the best place to introduce this level of multi-dimensional material survey (this is an editorial call). There are specialty journals in this sub-discipline (e.g. Journal of Biodegradation and Biodeterioration, Built Environment, etc.)

There are many authors on this manuscript; most this reviewer recognizes for specialty skills related analyses of environmental samples (e.g. microscopy, MS, bioinformatics) from the Built Environment (BE). This reviewer recognizes the “greater” design of this study is noble, clever and warranted: time-series of samples are recovered from a relatively large cohort of building material coupons (with actual and pseudo-replicates), which are “aliquoted”, split and analyzed for different biomarkers (biochemistry, PCR, genomics) in parallel. The authors then use advanced statistical analyses to discern correlations and co-occurrences of biomarkers, with different resolution. A tricky task to say the least, when managing different sampling and analytical needs that must “tie-together” otherwise very disparate skill sets. Therein lies the challenge and value of this work.

This reviewer found this work to be largely observational, leveraging statistical outcomes, through a broad range of significance levels, to implicate some generalized conditions and support conclusion. Some of these conditional conclusions appeared tangential and detracted from the core findings (mission) of the study (listed below).

Organization and Readability

This reviewer found this manuscript to be well written; it followed a logical path with coherent organization; however, it was voluminous and tedious to get through (considering the supplementary materials). In terms of readability, the relatively large cohort of statistical tests applied, with the range of significance thresholds, made it difficult to follow at times. I am not sure how to suggest to condense this, except to eliminate some of the lesser important “suggestions” that had no direct biochemical/genomic data to support the stated implications as follows:

Page 9, ln 362-365 biofilm formation (biofilm/carbohydrates not measured);

We have edited this section as suggested, and have reserved the majority of biofilm-related interpretations to the discussion section.

Page 9, line 345, suggestion of generalized alkaloid production by the Ascomycota (alkaloid pools not measured);

The original statement was intended to refer specifically to the alkaloids Nigragillin and Fumigaclavine C, not to generic alkaloids. We have clarified the text to reflect this.

Page 9, indication of the presence of R-azole-resistant *Penicillium* species (this ARG not measured), etc.)

Thiabendazole resistance is well known in the literature. We agree that specific detection of the resistance gene would provide corroboration, but unfortunately the 16S data do not facilitate this. We believe our data on their own still provide significant support for this hypothesis.

Page 10 *Pseudomonas* spp. specific dose-dependent response to antibacterial metabolites (S11).

We see this phrasing may be misleading and we have edited the text to remove the reference to dose-dependence.

If the manuscript contents were to be condensed to the context of its title, this reviewer would recommend completely dropping the virus observations, as they detract from the core genomics and metabolite findings (the salient theme of the work). While the influence of viruses “could’ be important, there was no data acquired which implicate how viruses may be associated with the microbial community dynamics and selected metabolomics observed.....just VLP counts. This reviewer believes the Prussin et al (2015) report was cited out of context (pg. 4, ln 128) since this pertains to VLP observations from PM recovered from filtered aerosol samples, and not the progression of microbial growth on surfaces (very different in BE context).

We agree that our original citation of Prussin *et al* was misleading, and we have now clarified that that paper focused on aerosol samples. Because so little work has yet been done to characterize VLP counts in the built environment, we have elected to keep that data in the revised manuscript even though it is slightly outside the main focus of the paper, as it may help motivate future studies.

Along the same theme, the photographs and generic description of generalize microbial growth (“visible” surface coverage) on the different dry/wet material types does not add scientific value to this manuscript. Visible mold does grow on wetted materials faster than their drier counter parts, and industrial hygienists routinely use visual confirmation of mold growth for building assessments. The photos and accompanying text add sensation and length to the text; but, in this reviewer’s opinion, really add nothing toward the core scientific contributions of the manuscript.

We agree that this section of our discussion falls outside the main focus of the paper, but we believe that data is useful for an intuitive understanding of the work as most readers will associate wetted materials with visible microbial growth, and may wonder about the extent to which we could changes in the community visually.

While potable water leaks are an appropriate representation for wetting with water source, this reviewer does not believe the water used to wet/saturate the material coupons is in anyway representative of flood-associated waters (abstract and introduction) and their potential as co-inoculums in this context. This reviewer suggests any consideration for this representation (flood waters) should be removed from the manuscript

We agree that the term “flood” may be too ambiguous. It can mean flooding due to a potable water leak, or it can mean flooding due to non-potable water (e.g. flood waters from outside, sewage, etc.). Given this ambiguity, we have modified the language to refer to more general terms (e.g. wetting by potable water or simulate a potable water flooding event).

Specific Technical Concerns

This reviewer presents the following technical concerns in the order of what I consider the most important in terms of supporting the work.

First and foremost is clarity with regard to the quantitative recovery of genetic materials from the different material surfaces, and the ability to reproducibly sequence these materials with high fidelity and minimal bias. What concerns me here is the description of broad spectrum quantitative PCR results as juxtaposed to the direct microscopy results (pg 4, lines 131 through 139). If BLP counts and broad spectrum qPCR were not in reasonable agreement on the different materials, this uncertainty lessens confidence that subsequent sequencing may not accurately represent the community abundance observations on which much of the manuscript is based. While the authors state on pg 4 line 139 that the different methods have different biases, there is no resolution to sequencing uncertainties introduced here (are they systematic biases at least?)

This effort was not part of the intended goal of the study, and yet is a valid concern. So while outside of the scope of the existing study, and while there are studies that try and deal with this concern, we acknowledge that future work needs to be done to address this.

In the context of comparing microscopy to qPCR, the authors present only BLP counts. Given the clear importance of fungi growing in and on wetted building materials, why are fungal spore counts absent from this report? Given VLP and BLP are included, and (much) more difficult to enumerate with direct microscopy than their fungal counterparts, why was the parallel juxtaposition of ITS qPCR and fungal spore counts absent?

The lack of fungal spore counts is an unfortunate limitation that reflects the biases toward some analytical methods versus others used by project team members. In retrospect, instead of focusing almost exclusively on molecular techniques, the addition of fungal spore counts and potentially hyphae length would have made a nice addition that ties the work back to more conventional approaches. However, we cannot retrospectively do this.

Acknowledging that the authors used a defined yet generic swabbing protocol to recover microbial samples, and applied some statistical rubric to define "reproducible" recovery, there is no discussion or acknowledgement regarding efforts to carefully standardize surface sampling. This is not a trivial issue and associated effort has been previously devoted to this topic in recent peer reviewed literature.

We appreciate the reviewer bringing up this point. We indeed thought a lot about how to approach sample collection, factoring in recovery efficiency, repeatability, and how much disturbance would be caused to our coupons by various techniques. We considered roller samplers, wipes, and other types of swabs, and settled on the polyester swabbing protocol we used. We likely traded off some collection efficiency and repeatability compared to other methods, but because many coupons got resampled, we weighed the advantages of small surface area and non-wetted swabbing more heavily because they would minimize disturbance on repeatedly sampled coupons than these other methods. We also added large numbers of replicates to further reduce these potential influences. We have now included more of discussion and a few citations that helped guide us to our decision.

Were the UV-ETOH sterilization protocols successful? Generic sterilization refers to a culture-based endpoint. The material coupons were not rendered DNA free (as judged by the analytical protocols, it does not appear so)? If this protocol did not, was the DNA baseline recovered consistent across the respective materials. If the authors could please expand this point and formally qualify sterilization where introduced in the methods section, pg 12, ln 490.

The reviewer is correct regarding culture-based endpoints, which we did not test. Unfortunately, we did not quantify the DNA recovered right after sterilization. We know from past experience that this method does not render surfaces DNA free. Moreover, we believe that even some microbial life from the interior recesses of the materials could survive sterilization and this community will potentially could come back to the material surface and interact with the microorganisms naturally inoculated from air, affecting the community composition that gets established on each different material. This was discussed on the *Experimental Setup* section and we have expanded the discussion in the methods section. This remains as a limitation to the protocol; however, the net effects should be small/negligible because the remaining DNA would be rendered non-viable. Moreover, for practical reasons, we had considered not sterilizing at all to better capture closer-to-real-life conditions in which building materials are not installed sterilized but rather are introduced with whatever microbial consortia persist after manufacturing shipping, storage, and install. So, the practical impacts of this limitation are not critical to our overall goals.

Reviewer #3

The manuscript by Lax et al investigates the dynamics of bacterial and fungal communities (and their metabolites) evolving over four surfaces of building materials found (presumably) in indoor environments.

Although the experiment is complex, the generated dataset complete and the analysis extensive, the specific questions and hypotheses leading this study are unclear. The authors had selected four materials, two of which are organic, but this is not fully exploited in the analysis or in the interpretation of the data. In addition, measurements characterizing the effect of biodeterioration on the physical properties of the materials selected are entirely absent.

Moreover, the experimental design includes incubation at an extremely high relative humidity over a long period of time, which is not necessarily compatible with environmental conditions in which biodeterioration could occur. The authors should clarify all these points as part of the introduction and justification of the study.

It is true that long-term high RH (94%) is not necessarily a realistic condition that a building material is exposed to. However, we chose it (1) to encourage maximum growth, which was more important to us than a realistic cycling of environmental conditions, and (2) much of the existing literature has used constant high RH conditions as well (e.g. Laks et al. 2002; Nielson et al. 2004; Hoang et al. 2010), with only limited studies investigating cyclic conditions (Johansson 2014; Johansson et al. 2013). We have modified the introductory language to explicitly state this limitation.

Specific comments:

Line 4-5. Indicating rRNA is misleading as the authors used DNA and not RNA for this analysis. This reviewer initially thought the authors investigated active microbial communities but later understood this was not the case.

We have altered the text to read “amplicon sequencing of the genes encoding 16S and ITS rRNA”

Line 71-72. The justification for the choice of the materials, as well as a more thorough description of those should be included.

The materials were selected as a sample of convenience to represent a relatively wide variety of common material types that were likely to experience high variability in microbial growth. We have now mentioned this in the paper.

Lines 91-114. The entire section is very difficult to understand. The section is titled “experimental design” but it is impossible to grasp the meaning of controls, the duration of the experiment, sterilization procedure, analysis performed on the surfaces (other than “looked similar”). All these details are indicated as part of the methods section, but they should be at least indicated briefly if the section is to be maintained. Moreover, the rationale behind the experiment and the hypothesis leading the experiment should be indicated.

We agree with the reviewer, and have made an effort to clarify the duration of the experiment and the inoculating locations in the first paragraph of the results section.

Line 199. Please select one nomenclature “coupons” or “tiles” but do not use those simultaneously unless there is different meaning attached to either one.

Agreed. We have selected “coupons” to avoid confusion and have made changes throughout.

Lines 502-504. Key environmental information such as spore load at the “inoculation location” should be provided. Also, it is unclear how the data obtained from the sampling sites was integrated in the analysis.

We assume that by “spore load” you are referring to the airborne concentration in each environment? We did not conduct air sampling in the inoculation locations. Although it could yield potentially helpful “input” data for interpreting our results, unfortunately cannot retroactively do so. The other sampling site data (e.g. temperature, RH, and occupant proximity) were metadata we collected did not vary much and thus are not expected to meaningfully affect the results.

Line 507. Please consider clarifying exactly the handling of the control group. What is the source of microorganisms for the wetted control material? Is it deposition? This is not clear.

The control group, i.e. the lab location, coupons were kept covered by aluminum foil in the lab. Therefore, deposition should have been minimal. Potential microbial sources include: tap water, microbes within the material matrix just beneath the surface, and DNA leftover from the sterilization process.

Line 522. Please indicate clearly the timing of the sampling points. T0-T6 is not informative and the reader needs to guess the experiment post-wetting lasted around 30 days.

We agree and have now made clear throughout the manuscript how many days post-incubation each timepoint represents.

Reviewers' Comments:

Reviewer #1:

Remarks to the Author:

The authors have responded to comments and for the most part addressed the suggestions. There are a few minor editing clarifications that need to be made:

Figures note "days for inoculation", while the text still reads T0 to T6.

Figure 1: Bacteria Count and BLP used interchangeably but should be consistent.

Were the fungal amplicons 2x250 (as written in response) or 2x300 (as written in text)?

How were OTUs generated from raw sequences for both bacteria and fungi?

I do wonder about how wide the audience is for work of this sort, but I do think the study is robust.

Additional comments regarding Reviewer 2's technical concerns:

Reviewer 2 comment - "First and foremost is clarity with regard to the quantitative recovery of genetic materials from the different material surfaces, and the ability to reproducibly sequence these materials with high fidelity and minimal bias. What concerns me here is the description of broad spectrum quantitative PCR results as juxtaposed to the direct microscopy results (pg 4, lines 131 through 139). If BLP counts and broad spectrum qPCR were not in reasonable agreement on the different materials, this uncertainty lessens confidence that subsequent sequencing may not accurately represent the community abundance observations on which much of the manuscript is based. While the authors state on pg 4 line 139 that the different methods have different biases, there is no resolution to sequencing uncertainties introduced here (are they systematic biases at least?)"

Reviewer 1 suggestion - The authors acknowledge the concern in their response but do not take efforts to address the concern in the manuscript. As I see it, the reviewer is asking the study authors to consider how some of the acknowledged potential biases regarding material recovery and analytical tools might influence their findings, which the authors could do as a point of discussion. For example, given the lack of agreement between BLP counts and broad spectrum qPCR, is there a reason to trust one more than the other? And if it's not the sequence-based approach, how might that affect the community abundance observations that underlie the bulk of the paper? Which biases are unique to each approach and which are systematic?

Reviewer 2 comment - "In the context of comparing microscopy to qPCR, the authors present only BLP counts. Given the clear importance of fungi growing in and on wetted building materials, why are fungal spore counts absent from this report? Given VLP and BLP are included, and (much) more difficult to enumerate with direct microscopy than their fungal counterparts, why was the parallel juxtaposition of ITS qPCR and fungal spore counts absent?"

Reviewer 1 suggestion - Perhaps mentioning in the text that counting fungal components would have been a valuable addition but are not included in this particular study would address the reviewer concerns.

Reviewer 2 comment - "Acknowledging that the authors used a defined yet generic swabbing protocol to recover microbial samples, and applied some statistical rubric to define "reproducible" recovery, there is no discussion or acknowledgement regarding efforts to carefully standardize surface sampling. This is not a trivial issue and associated effort has been previously devoted to this topic in recent peer reviewed literature."

Reviewer 1 suggestion - As with the point above, acknowledging how sample recovery might affect conclusions in the paper would address reviewer concerns.

Reviewer 2 comment - "Were the UV-ETOH sterilization protocols successful? Generic sterilization refers to a culture-based endpoint. The material coupons were not rendered DNA free (as judged by the analytical protocols, it does not appear so)? If this protocol did not, was the DNA baseline recovered consistent across the respective materials. If the authors could please expand this point and formally qualify sterilization where introduced in the methods section, pg 12, ln 490."

Reviewer 1 suggestion - The manuscript now acknowledges that the samples were not made DNA free, but does not address the reviewer concerns about the consistency of this "background" DNA and how it might affect conclusions. Negative controls of the samples prior to experimentation would have answered this question. In general including negative controls/blanks for microbiome samples is considered a good practice, and that doesn't seem to have been done in this study. Again, at this point the authors can only acknowledge in the manuscript that negatives were not included.

REVIEWERS' COMMENTS:

Reviewer #1 (Remarks to the Author):

The authors have responded to comments and for the most part addressed the suggestions. There are a few minor editing clarifications that need to be made:

Figures note "days for inoculation", while the text still reads T0 to T6.

We have now removed all mention of timepoints in the manuscript, and refer only to the number of days post-incubation.

Figure 1: Bacteria Count and BLP used interchangeably but should be consistent.

We now refer exclusively to BLP counts

Were the fungal amplicons 2x250 (as written in response) or 2x300 (as written in text)?

Thank you for catching this discrepancy. They were 2x250, as is now stated in the manuscript.

How were OTUs generated from raw sequences for both bacteria and fungi?

We have now added a description of our OTU clustering to the Methods section (beginning line 722)

I do wonder about how wide the audience is for work of this sort, but I do think the study is robust.

Additional comments regarding Reviewer 2's technical concerns:

Reviewer 2 comment - "First and foremost is clarity with regard to the quantitative recovery of genetic materials from the different material surfaces, and the ability to reproducibly sequence these materials with high fidelity and minimal bias. What concerns me here is the description of broad spectrum quantitative PCR results as juxtaposed to the direct microscopy results (pg 4, lines 131 through 139). If BLP counts and broad spectrum qPCR were not in reasonable agreement on the different materials, this uncertainty lessens confidence that subsequent sequencing may not accurately represent the community abundance observations on which much of the manuscript is based. While the authors state on pg 4 line 139 that the different methods have different biases, there is no resolution to sequencing uncertainties introduced here (are they systematic biases at least?)"

Reviewer 1 suggestion - The authors acknowledge the concern in their response but do not take efforts to address the concern in the manuscript. As I see it, the reviewer is asking the study authors to consider how some of the acknowledged potential biases regarding material recovery and analytical tools might influence their findings, which the authors could do as a point of discussion. For example, given the lack of agreement between BLP counts and broad spectrum qPCR, is there a reason to trust one more than the other? And if it's not the sequence-based approach, how might that affect the community abundance observations that underlie the bulk of the paper? Which biases are unique to each approach and which are systematic?

There are a number of factors which might explain this difference. In microscopy, error comes from swabbing and preparation, while in qPCR, error can originate from DNA extraction, the PCR reaction, PCR primers not binding equally to all sequences, and 16S copy number. We have added this text to our Methods section (lines 700-703).

Reviewer 2 comment - "In the context of comparing microscopy to qPCR, the authors present only BLP counts. Given the clear importance of fungi growing in and on wetted building materials, why are fungal spore counts absent from this report? Given VLP and BLP are included, and (much) more difficult to enumerate with direct microscopy than their fungal counterparts, why was the parallel juxtaposition of ITS qPCR and fungal spore counts absent?"

Reviewer 1 suggestion - Perhaps mentioning in the text that counting fungal components would have been a valuable addition but are not included in this particular study would address the reviewer concerns.

We decided against counting fungal spores because of hyphae which are multicellular and very difficult to estimate counts of. We have added this explanation to the manuscript (lines 646-647).

Reviewer 2 comment - "Acknowledging that the authors used a defined yet generic swabbing protocol to recover microbial samples, and applied some statistical rubric to define "reproducible" recovery, there is no discussion or acknowledgement regarding efforts to carefully standardize surface sampling. This is not a trivial issue and associated effort has been previously devoted to this topic in recent peer reviewed literature."

Reviewer 1 suggestion - As with the point above, acknowledging how sample recovery might affect conclusions in the paper would address reviewer concerns.

We appreciate the reviewer bringing up this point. We indeed thought a lot about how to approach sample collection, factoring in recovery efficiency, repeatability, and how much disturbance would be caused to our coupons. We considered roller samplers, wipes, and other types of swabs, and settled on the polyester swabbing protocol we used. We likely traded off some collection efficiency and repeatability compared to other methods, but because many coupons got resampled, we weighed the advantages of small surface area and non-wetted swabbing more heavily because they would minimize disturbance on repeatedly sampled coupons than these other methods. We also added large numbers of replicates to further reduce these potential influences. We have now included more of discussion and a few citations that helped guide us to our decision (lines 604-617)

Reviewer 2 comment - "Were the UV-ETOH sterilization protocols successful? Generic sterilization refers to a culture-based endpoint. The material coupons were not rendered DNA free (as judged by the analytical protocols, it does not appear so)? If this protocol did not, was the DNA baseline recovered consistent across the respective materials. If the authors could please expand this point and formally qualify sterilization where introduced in the methods section, pg 12, ln 490."

Reviewer 1 suggestion - The manuscript now acknowledges that the samples were not made DNA free, but does not address the reviewer concerns about the consistency of this "background" DNA and how it might affect conclusions. Negative controls of the samples prior to experimentation would have answered this question. In general including negative controls/blanks for microbiome samples is considered a good practice, and that doesn't seem to have been done in this study. Again, at this point the authors can only acknowledge in the manuscript that negatives were not included.

We agree that sampling each tile before inoculation to assess the "background DNA" of the materials would have been advisable, and have added text specifying this (line 617-618).